# A CRISPR-based ultrasensitive assay detects attomolar concentrations of SARS-CoV-2 antibodies in clinical samples

Yanan Tang[1,7], Turun Song[2,7], Lu Gao[1], Saifu Yin[2], Ming Ma[2], Yun Tan[1], Lijuan Wu[3], Yang Yang[4], Yanqun Wang[5], Tao Lin ©[2] ✉ & Feng Li ©[1,6] ✉

CRISPR diagnostics are powerful tools for detecting nucleic acids but are generally not deployable for the detection of clinically important proteins. Here, we report an ultrasensitive CRISPR-based antibody detection (UCAD) assay that translates the detection of anti-SARS-CoV-2 antibodies into CRISPR-based nucleic acid detection in a homogeneous solution and is 10,000 times more sensitive than the classic immunoassays. Clinical validation using serum samples collected from the general population (n = 197), demonstrates that UCAD has 100% sensitivity and 98.5% specificity. With ultrahigh sensitivity, UCAD enables the quantitative analysis of serum anti-SARS-CoV-2 levels in vaccinated kidney transplant recipients who are shown to produce "unde-tectable" anti-SARS-CoV-2 using standard immunoassay. Because of the high sensitivity and simplicity, we anticipate that, upon further clinical validation against large cohorts of clinical samples, UCAD will find wide applications for clinical uses in both centralized laboratories and point-of-care settings.

Coronavirus disease 2019 (COVID-19), caused by severe acute respiratory syndrome coronavirus 2 (SARS-CoV-2) was declared a pandemic by the World Health Organization (WHO) on 11 March 2020[1]. The urgent requirements for sensitive and specific assays to detect viral infections and subsequent immune responses at both individual and population levels offer tremendous opportunities for accelerating the development and validation of new diagnostic techniques for clinical uses. Clustered regularly interspaced short palindromic repeats (CRISPR)-based point-of-care tests (POCTs) are among the most attractive tools for diagnosing COVID-19 infections via the detection of viral RNA under resource-limited conditions[2,3]. Currently, both CRISPR-Cas13 specific high-sensitivity enzymatic reporter unlocking (SHERLOCK) and Cas12-mediated DNA endonuclease-targeted CRISPR trans reporter (DETECR) have been approved by the United States Food and Drug Administration (FDA) for commercial uses through Emergency Use Authorization (EUA)[3-5]. Despite remarkable success as nucleic acid tests (NATs), CRISPR-Cas systems have not been exploited for the sensitive detection of anti-SARS-CoV-2 antibodies, another important class of biomarkers for diagnosis, estimating the seroprevalence of infection, monitoring the progress of pandemic, and assessing the protective humoral immunity of infected individuals and recipients of vaccination[6-8].

To date, many immunoassays, such as enzyme-linked immuno-sorbent assay (ELISA) and chemiluminescence immunoassay (CLIA), have been developed and commercialized for detecting anti-SARS-CoV-2 antibodies[9-12] but are generally not optimal for uses as POCT due

[1]Key Laboratory of Green Chemistry & Technology of Ministry of Education, College of Chemistry, Analytical & Testing Center, Sichuan University, Chengdu, Sichuan 610064, China. [2]Urology Department, Urology Research Institute, Organ Transplantation Center, West China Hospital, Sichuan University, Chengdu, Sichuan 610041, China. [3]Department of Laboratory Medicine, West China Hospital, Sichuan University, Chengdu, Sichuan 610041, China. [4]Shenzhen Key Laboratory of Pathogen and Immunity, Shenzhen Third People's Hospital, Second Hospital Affiliated to Southern University of Science and Technology, Shenzhen, China. [5]State Key Laboratory of Respiratory Disease, National Clinical Research Centre for Respiratory Disease, Guangzhou Institute of Respiratory Health, the First Affiliated Hospital of Guangzhou Medical University, Guangzhou, Guangdong, China. [6]Department of Chemistry, Centre for Biotechnology, Brock University, St. Catharines, Ontario, ON L2S 3A1, Canada. [7]These authors contributed equally: Yanan Tang, Turun Song. ✉e-mail: kidney5@163.com; windtalker_1205@scu.edu.cn

to tedious washing and operational steps[13,14]. More importantly, these conventional assays lack the sensitivity to detect antibodies at the early stages of infections or to monitor changes in anti-SARS-CoV-2 levels in immunocompromised patients who are subject to much-reduced seroconversion upon COVID-19 infection or vaccination[7,12,15–18]. Therefore, a simple and ultrasensitive antibody test with limit-of-detection (LOD) comparable to those of CRISPR-based NATs will not only enable early diagnosis of COVID-19 upon the onset of symptoms but also shed light on humoral immune responses to infection or vaccination for immunocompromised subpopulations who produce "undetectable" anti-SARS-CoV-2 antibodies defined by conventional techniques.

Herein, we introduce an ultrasensitive CRISPR-based antibody detection (UCAD) assay that is 10,000 times more sensitive than the commercial ELISA kits for the detection of anti-SARS-CoV-2 spike protein receptor binding domain (anti-RBD) IgG and IgM in undiluted human serum samples. UCAD is also highly specific and modular. It can effectively distinguish anti-SARS-CoV-2 from antibodies against other coronaviruses, such as SARS-CoV and MERS-CoV. UCAD can also be engineered to respond specifically to antibodies produced against the wild-type (WT) SARS-CoV-2, Delta, or Omicron mutants, by simply switching the RBD motif for target recognition. The detection module of UCAD is also switchable between fluorescence and a lateral flow visual readout, making it particularly advantageous for diagnosis in POC settings. UCAD is clinically validated using 65 anti-SARS-CoV-2 positive serum samples collected from healthy participants after two doses of inactivated COVID-19 vaccines as positive controls and 77 human serum samples collected before the pandemic as negative controls. The specificity of UCAD is further validated against 55 clinically identified COVID-19 negative serum samples collected during the pandemic. We finally deploy UCAD for the ultrasensitive detection of anti-RBD IgG and IgM in clinical serum samples collected from a cohort of 85 kidney transplant receivers (KTRs) who received two doses of inactivated COVID-19 vaccine. Because of the ultrahigh sensitivity, UCAD allows the detection of anti-SARS-CoV-2 levels in this representative immunocompromised subpopulation whose antibody levels were defined as "undetectable" by standard CLIA. Finally, we show that UCAD can detect increasing levels of anti-RBD IgG in 28 out of 33 KTRs who received the third dose of COVID-19 vaccination.

## Results

### Design and operation of UCAD assay

The idea of UCAD is to convert the detection of anti-SARS-CoV-2 into the production of a predesigned CRISPR-Cas12a targetable double-stranded DNA (dsDNA) barcode, so that the recombinase polymerase amplification (RPA) and subsequent indiscriminate single-stranded DNase (ssDNase) activity of Cas12a can be unleashed by the antibody (Fig. 1a). To achieve this goal, we first designed a dsDNA barcode containing a 20-nucleotide (nt) binding domain to the CRISPR guide RNA (crRNA) (Fig. 1b). To disrupt the binding and activation of Cas12a in the absence of the antibody, we truncated the dsDNA barcode at both the target strand (TS) and nontarget strand (NTS) at the middle of the crRNA binding sequence. The TS and NTS probes were then conjugated with the SARS-CoV-2 spike protein RBD and an anti-human IgG (or IgM) antibody, respectively (Fig. 1c). A short complementary domain was left between the two probes to initiate antibody-induced proximity hybridization (Fig. 1c). Driven by a T4 polymerase, the complete dsDNA barcode forms via primer extension, which serves as a surrogate for the amplified detection of anti-RBD. Experimentally, the optimal length of the complementary domain between the TS and NTS probes was determined to be 6 nt (Supplementary Fig. 1). With this design, the melting temperature ($T_m$) was estimated to be only ~10 °C, so that no stable duplex could be formed at 37 °C (Fig. 1c). In the presence of anti-RBD, the binding of anti-human IgG (or IgM) and RBD to the same antibody brings the two DNA probes into proximity and

thus leads to the formation of a stable duplex with an estimated $T_m$ of 46 °C (Fig. 1c).

The assay protocol of UCAD consists of three simple steps (Fig. 1a), including (1) antibody-specific primer extension to produce dsDNA barcodes; (2) RPA amplification; and (3) the cleavage of fluorophore-quencher (FQ) labeled ssDNA reporters mediated by CRISPR-Cas12a. Reactions at each step were all performed homogenously at a constant 37 °C in a single test tube without the need for any separation steps. By carefully optimizing the assay conditions (Supplementary Figs. 2, 3), we were able to detect as low as 10 aM anti-RBD human monoclonal antibody, a LOD comparable to those of CRISPR-based NATs (Fig. 1f and Supplementary Fig. 4). Further kinetic analysis suggested that 40 min was optimal for endpoint fluorescence detection, as it offered the best dynamic range for quantitative analysis (Supplementary Fig. 4d). The high sensitivity and low background of UCAD were also confirmed using polyacrylamide gel electrophoresis (PAGE) analysis (Supplementary Fig. 5).

We next examined whether UCAD would be effective for detecting anti-RBD IgG and IgM in human serum samples. The kinetic curves in Fig. 1d, e demonstrate that UCAD could effectively detect both anti-RBD IgG and IgM in certified IgG/IgM positive human serum samples obtained commercially, whereas very little fluorescence increases were shown in the negative samples. The sensitivity of UCAD was further compared with standard ELISA using a series of human serum samples prepared by diluting the IgG/IgM positive serum using the negative serum with dilution factors ranging from 1 to 100,000. UCAD was able to detect anti-RBD IgG (Fig. 1g and Supplementary Fig. 6a) and IgM (Fig. 1h and Supplementary Fig. 6b) throughout the dilution series. In contrast, the standard ELISA kit worked only for ten fold diluted serum samples (Fig. 1g, h), suggesting that our UCAD assay is at least 10,000 more sensitive than ELISA.

### Specificity and modularity of UCAD

UCAD was also determined to be highly specific, evidenced by the low cross-reactivity to human monoclonal antibodies against MERS-CoV spike protein RBD and SARS-CoV-2 nucleocapsid (N) protein, as well as rabbit polyclonal antibodies against SARS-CoV-2 throughout concentration ranges from 10 aM to 1 pM (Fig. 2a and Supplementary Fig. 7). The specificity of UCAD was further validated against three anti-SARS-CoV positive human serum samples (Supplementary Fig. 8). Results in Fig. 2b, c revealed background level of fluorescence in SARS-CoV serum samples (ANOVA, $p = 0.0676$ for IgG, $p = 0.1940$ for IgM), confirming the low cross-reactivity of UCAD against anti-SARS-CoV in clinical serum samples.

UCAD is also highly modulable, the specificity of which can be engineered by simply switching the antibody-recognition domain conjugated with the TS probe (Fig. 2d). To demonstrate the modularity, we first modified TS probes with RBDs from the wild-type (WT) SARS-CoV-2, Delta mutant, and Omicron mutant, respectively, and validated the UCAD assay against monoclonal antibodies produced by the three subtypes of SARS-CoV-2 (Fig. 2e and Supplementary Fig. 9). Excitingly, highest fluorescence signals were observed for all three types of anti-RBD antibodies only when correct RBD domain was employed for the UCAD assay throughout a concentration range from 100 aM to 1 pM. Moderate cross-reactivities were observed in Fig. 2e and Supplementary Fig. 9, likely because the conservative domains remained at the mutated RBDs, which was further confirmed by the lowest level of cross-reactivity between WT and Omicron. It is also possible to engineer the UCAD for the specific detection of anti-SARS-CoV-2 N protein by switching the RBD motif into the N protein (Fig. 2d). Results in Fig. 2f and Supplementary Fig. 9d confirmed the high sensitivity for the detection of anti-SARS-CoV-2 N protein and low cross-reactivity with anti-RBD antibodies.

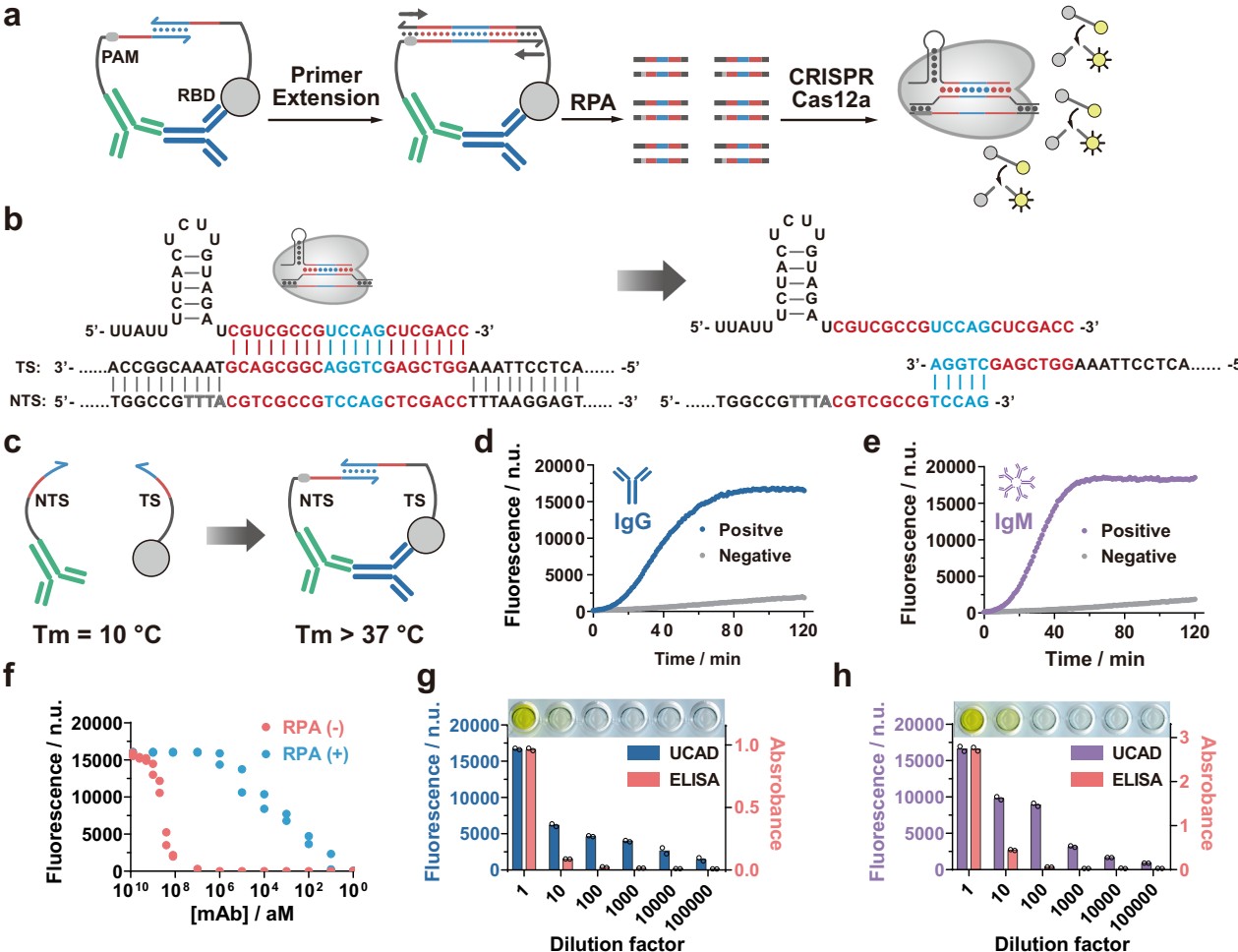

**Fig. 1 | Design principle and sensitivity of UCAD. a** Schematic illustration of the workflow of the UCAD assay. **b** Sequence design of the dsDNA template for triggering the collateral cleavage activity of CRISPR-Cas12a, as well as the strategy to split the dsDNA template into TS and NTS probes for UCAD. **c** Proximity binding of the two UCAD probes to the same anti-RBD antibody through affinity motifs enhances the T_m of the duplex domain from 10 to 46 °C and thus enables the stable binding between TS and NTS probes. **d**, **e** Kinetic curves for the detection of anti-RBD IgG (**d**) and IgM (**e**) in commercially purchased certified anti-SARS-CoV-2 IgG/IgM positive and negative serum samples using UCAD. **f** A calibration curve for the detection of anti-SARS-CoV-2 spike protein RBD human mAb (clone: OTIH401) in the range of 1 aM to 1 pM was established by plotting fluorescence intensity at 40 min of the CRISPR-Cas12a reaction as a function of target concentrations, which was also compared with UCAD without RPA amplification. Each concentration has replicated measurements ($n = 2$). **g**, **h** Detection of anti-RBD IgG (**g**) and IgM (**h**) in certified anti-SARS-CoV-2 IgG/IgM positive serum diluted in negative human serum with dilution factors from 1 to 100,000 using both UCAD and commercial ELISA kits. Each concentration has replicated measurements ($n = 2$). Source data are available in the Source Data file.

## Clinical validation of UCAD

We validated UCAD against 65 positive clinical human sera that were previously tested by the West China Hospital of Sichuan University using a standard total anti-RBD CLIA assay (female: $n = 37$, 56.9%; male: $n = 28$, 43.1%; age: 45.5 ± 17.1) and 77 pre-pandemic sera collected in 2019 before the outbreak of COVID-19. All CLIA-positive samples were collected from healthy participants who had two doses of inactivated COVID-19 vaccine (Supplementary Data 1) and their IgG and IgM levels against RBD were measured by UCAD without selection (Fig. 3b, c). Significant differences were found between the positive and pre-pandemic cohorts ($p < 0.0001$, $n = 142$) (Fig. 3d, e). To further evaluate the clinical specificity of UCAD, we also obtained a set of sera collected from 55 unvaccinated healthy participants during the pandemic (female: $n = 12$, 21.8%; male: $n = 43$, 78.2%; age: 30.2 ± 7.5), who have been confirmed to be anti-RBD negative using the standard CLIA test (Supplementary Data 2). Comparing with the pre-pandemic cohorts, neither IgG nor IgM levels were significantly different in this cohort (Fig. 3d, e). The 55 negative sera were combined with the 77 pre-pandemic sera as a new negative group to determine the optimal cutoff values for serum anti-RBD IgG and IgM via receiver operating

characteristics (ROC) curve analysis (Fig. 3f). The binary outcomes (positive or negative) for anti-RBD IgG and IgM were then combined, where either IgG or IgM positive was identified as a positive instance. The test results by UCAD were then compared with the total antibody test by the standard CLIA using a confusion matrix (Fig. 3g). The sensitivity and specificity of UCAD ($n = 197$) were determined to be 100 and 98.5%, respectively (Fig. 3g). Altogether, within the total 132 pre-pandemic and CLIA-confirmed negative samples, we only found two instances where UCAD were not in agreement with CLIA. As both samples were found in the cohort of the 55 post-pandemic CLIA negative sera, there is a possibility that the two participants had been exposed to viruses and produced a low level of anti-RBD below the LOD of CLIA.

## Detection of anti-RBD IgG/IgM in vaccinated KTRs

To further demonstrate the clinical potential of UCAD to detect low abundant antibodies in immunocompromised population, we studied a cohort of 85 KTRs who had received 2 doses of inactivated COVID-19 vaccine using UCAD (female: $n = 24$, 28.2%; male: $n = 61$, 71.8%; age: 42.3 ± 9.7, Fig. 4a). All KTRs received a triad

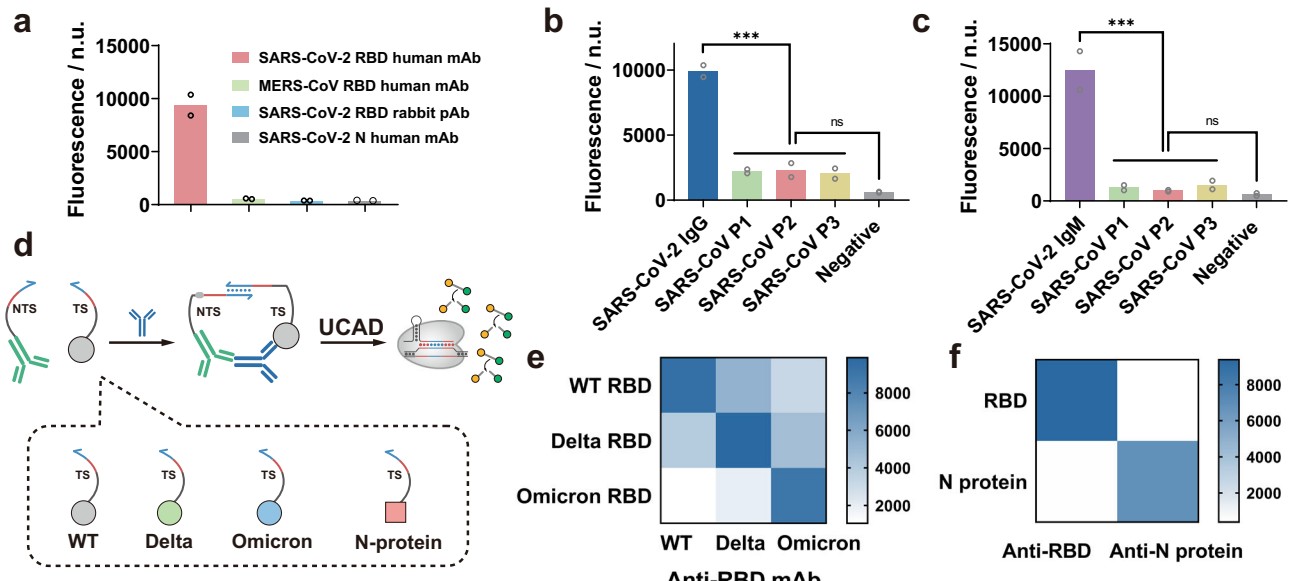

**Fig. 2 | Specificity and modularity of UCAD. a** Detection of the target anti-human RBD and closely related nontargeted anti-MERS-CoV spike protein RBD human mAb (clone: m336), anti-SARS-CoV-2 nucleocapsid (N) protein human mAb, and anti-SARS-CoV-2 spike protein RBD rabbit pAb at a concentration of 10 fM using UCAD. Each sample was measured twice in two independent experiments. **b, c** Evaluation of the specificity of UCAD for anti-RBD IgG (**b**) and IgM (**c**) against anti-SARS-CoV in clinical sera from three SARS patients in 2003. Each serum sample was measured twice in two independent experiments. The UCAD signals of anti-SARS-CoV positive sera were significantly lower than the anti-SARS-CoV-2 positive serum ($p = 0.0004$ for IgG, $p = 0.0025$ for IgM). Ordinary one-way ANOVA were used to compare the difference between multiple groups. **d** Schematic illustration of the modularity of UCAD for varying mutants of RBD and the N protein of SARS-CoV-2 by switching the recognition motif on the TS probe. **e** Heatmap of the detection of 10 fM wild-type (WT), Delta specific, and Omicron specific anti-RBD human mAb by using WT RBD, Delta RBD, and Omicron RBD (B.1.1.529) modified TS probes. **f** Heatmap of the detection of 10 fM WT RBD human mAb and anti-N protein human mAb with WT RBD and N protein modified TS probes. Source data are available in the Source Data file. ns: $p > 0.05$, ***$p \leq 0.001$.

immunosuppression regimen composed of mycophenolate mofetil (500 mg bid), steroids (5–10 mg qd), and tacrolimus (TAC) with target trough levels of 5 to 8 ng/mL. More details on demographic and therapeutic information are listed in Supplementary Tables 2 and Supplementary Data 3. Sex and age were found to have no significant impact on the levels of anti-RBD IgG and IgM determined using UCAD (Supplementary Fig. 10).

Serum samples were collected 10 to 60 days (median = 26 days) after the second dose of vaccine. Consistent with previous reports[19], only five out of 85 KTRs (5.9%) were found to be anti-RBD positive using the standard CLIA assays (Supplementary Data 3). In contrast, a total of 73 KTRs (85.9%) were determined to be anti-RBD positive using UCAD, including 54 IgG+ and 67 IgM+ (Fig. 4b–e). The Venn diagrams in Fig. 4f revealed that 48 out of 85 KTRs were IgG+/IgM+ (56.5%), six were IgG+/IgM− (7.1%), 19 were IgG−/IgM+ (22.4%), and 12 were IgG−/IgM− (14.1%). For both anti-RBD IgG and IgM, UCAD signals of the KTR cohort were significantly higher than those of the negative controls ($n = 217$, $p < 0.0001$) (Fig. 4d, e). As expected, substantially reduced humoral immunity was observed in KTRs in response to vaccination, evidenced by the much lower levels of anti-RBD IgG and IgM than those of the vaccinated healthy cohort ($n = 150$, $P < 0.0001$) (Fig. 4d, e).

Upon UCAD analysis, we were able to classify the 85 KTRs into the anti-RBD positive group (IgG+/IgM+, IgG+/IgM−, or IgG−/IgM+, $n = 73$) and negative group (IgG−/IgM−, $n = 12$). A further flow cytometry analysis of the cellular immunities of KTR subgroups revealed a substantially lower level of plasmablasts in IgG−/IgM− KTRs (Supplementary Fig. 11b). As plasmablasts are a positive indicator for the induction of long-lived plasma cell responses upon COVID-19 vaccination[20], the observed low level of plasmablasts was well correlated with the undetectable level of anti-RBD in this group of KTRs. In addition to plasmablasts, the cellular analysis of CD4+ cells also showed significant differences in Th2 and Th17 cells in IgG+/IgM+ and IgG−/IgM− KTRs (Supplementary Fig. 11c, d), suggesting that the abnormality of B cell

differentiation and reduced antibody production in anti-RBD negative KTRs might also be related to T cell regulation[21–23].

We also employed UCAD to monitor the changes in anti-RBD IgG levels in 33 KTRs who received the third dose of inactivated COVID-19 vaccine. Sera were collected between 8–179 days (median = 94) after receiving the third dose of vaccine. All 33 KTRs were found to be anti-RBD IgG negative after the second vaccine dose and only two were found to turn positive after the third dose using the standard CLIA (Supplementary Data 3). By quantifying the "undetectable" levels of anti-RBD IgG using UCAD, we observed a significant elevation of anti-RBD IgG levels in 28 out of 33 (84.8%) KTRs after receiving the third dose with fold changes as much as 6.4 times (Fig. 4g, h and Supplementary Fig. 33).

**Integrating UCAD with lateral flow readout**

As a CRISPR-based ultrasensitive assay, UCAD also holds great promise as a POCT for detecting anti-SARS-CoV-2 antibodies under resource-limited conditions. We finally engineered a lateral flow readout for UCAD by replacing the fluorophore-quencher labeled ssDNA reporter with one labeled with FAM and digoxin (Dig) at the 3′ and 5′ ends, respectively (Fig. 5). A lateral flow test strip was constructed with anti-Dig immobilized at the control line (C-line) and anti-goat IgG at the test line (T-line) (Fig. 5b). In the absence of the target antibodies, the FAM-Dig labeled ssDNA reporter remains intact, which helps capture all goat anti-FAM labeled gold nanoparticles (AuNPs) at the C-line. In the presence of the anti-RBD IgG or IgM, the ssDNA reporters were cleaved by the target-induced collateral activity of Cas12a. As a result, AuNPs were captured at the T-line. We then tested UCAD with a lateral flow readout for the detection of anti-RBD IgG and IgM in clinical sera. Consistent with our fluorescence-based readout, we were able to clearly identify 15 anti-SARS-CoV-2 positive clinical sera, which were confirmed by both UCAD and CLIA from 15 pre-pandemic negative sera using the naked eye (Fig. 5c, d).

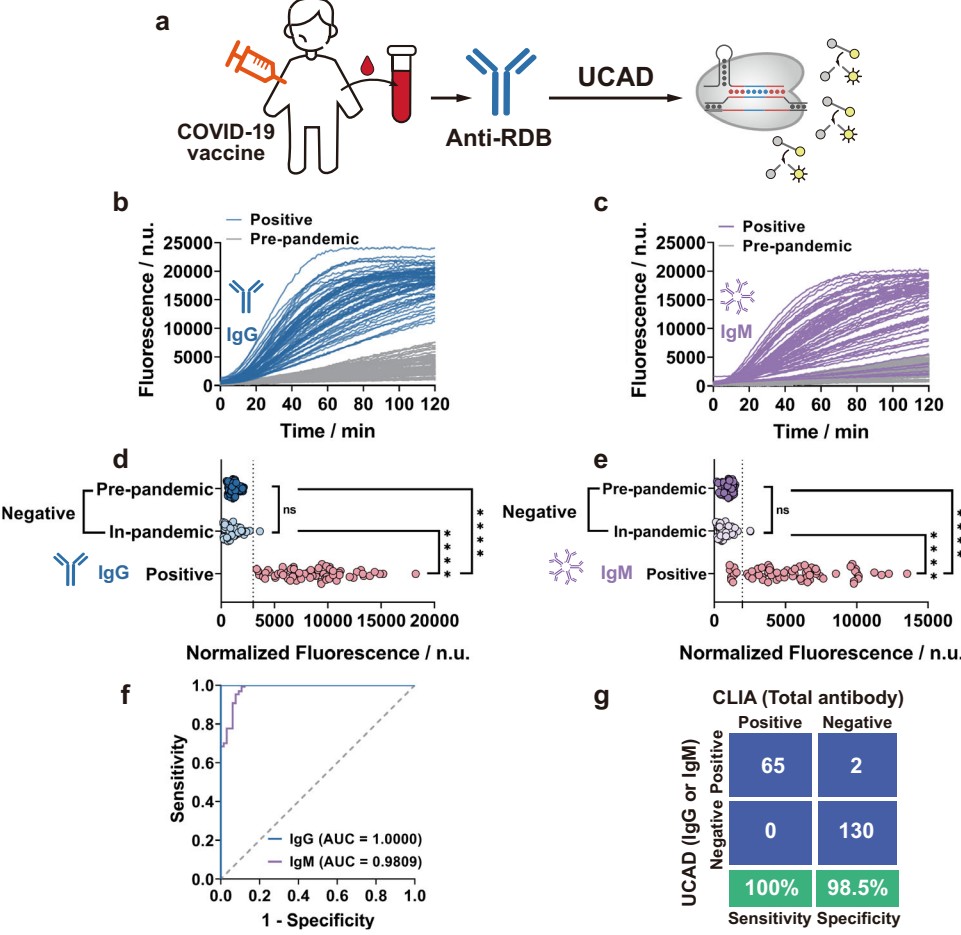

**Fig. 3 | Clinical validation of UCAD. a** Schematic illustration of the workflow for clinical validation of UCAD. **b, c** Kinetic curves for the detection of anti-RBD IgG (**b**) and IgM (**c**) in 65 anti-RBD positive sera collected from healthy individuals who received two shots of inactivated COVID-19 vaccine and 77 pre-pandemic sera. **d, e** Endpoint fluorescence signals at 40 min of three clinical cohorts, including 77 pre-pandemic samples, 65 positive samples, and 55 presumptive negative sera collected during the pandemic. Both the anti-RBD IgG and IgM levels of the positive cohort ($n = 65$) were significantly higher than those of the pre-pandemic ($p = 6.76e^{-27}$ for IgG and $p = 9.15e^{-15}$ for IgM) and in-pandemic negative cohorts ($p = 1.44e^{-20}$ for IgG and $p = 2.59e^{-18}$ for IgM). But the pre-pandemic and in-pandemic negative cohorts had no significant difference ($p = 0.2595$ for IgG and $p = 0.0607$ for IgM) and were thus combined as one

negative group ($n = 132$) to determine cutoffs for distinguishing positive and negative results in UCAD. Unpaired two-tailed t-tests were used to evaluate statistical differences between cohorts. **f** ROC curves of the UCAD assay for detecting anti-RBD IgG and IgM in 197 human serum samples (positive: $n = 65$, negative: $n = 132$). Optimal cutoff fluorescence values were selected through ROC analysis: 3007 n.u. at 40 min for anti-RBD IgG (specificity = 100%, sensitivity = 100%) and 1973 n.u. at 40 min for anti-RBD IgM (specificity = 87.69%, sensitivity = 100%). **g** Evaluation of UCAD sensitivity and specificity compared to standard CLIA test using a confusion matrix ($n = 132$ for negative sera and $n = 65$ for positive sera). The sensitivity of UCAD was 100% (95% confidence interval: 93.0–100%) and its specificity was 98.5% (95% confidence interval: 94.1–99.7%). Source data are available in the Source Data file. ns: $p > 0.05$, ****$p ≤ 0.0001$.

## Discussion

In this work, we have introduced UCAD technology that represents a technical advance and paradigm shift of CRISPR diagnostics from solely NATs to ultrasensitive protein analysis. The past few years have witnessed the remarkable success of CRISPR diagnostics as fast and cost-effective POC NATs are accessible for patients in low-resource settings. Because UCAD is an "add-on" to existing CRISPR diagnosis with a one-step conversion from protein to DNA barcode in a homogeneous solution, it is readily adaptable to existing detection platforms for SHERLOCK or DETECTR without compromising any sensitivity or simplicity. Here, we demonstrate that UCAD is 10,000 times more sensitive than standard ELISA for the detection of serum IgG and IgM against SARS-CoV-2 RBD. Comparing existing efforts to harness CRISPR systems for protein detection, UCAD demonstrates remarkable higher sensitivity and lower LOD not only in buffer systems but also in real clinical settings[24,25]. UCAD is also highly modulable, allowing the sensitive and specific detection of anti-SARS-CoV-2 N protein and antibodies against Delta and Omicron RBDs by switching the target recognition motif on the TS probe. We also show that the

fluorescent readout of UCAD could be readily converted to a lateral flow test with a visual readout, which was successfully validated using clinical sera. Because of the simple, homogeneous nature of UCAD, it is also possible to integrate UCAD with many other visual, colorimetric detection platforms, such as the use of plasmonic gold nanoparticles[26], for field-based applications, or with droplet microfluidic systems[27] to further enhance the sensitivity and quantification capacity. Compared to existing ultrasensitive protein detection techniques, such as single molecular array (Simoa)[28,29] or proximity ligation assays (PLA)[13,14,30], UCAD works isothermally at 37 °C without the need for any specialized equipment or tedious operation and washing steps and thus is particularly attractive for use in resource-constrained conditions, including the opportunity for self-monitoring. With the ultrahigh sensitivity, modularity, and simplicity, we anticipate that UCAD will find broad applications for early diagnosis and monitoring of emerging infectious diseases caused by coronaviruses, Zika virus, monkeypox virus, and beyond. It may also be engineered for detecting low abundant protein-based biomarkers that are critically related to somatic diseases, such as cancer.

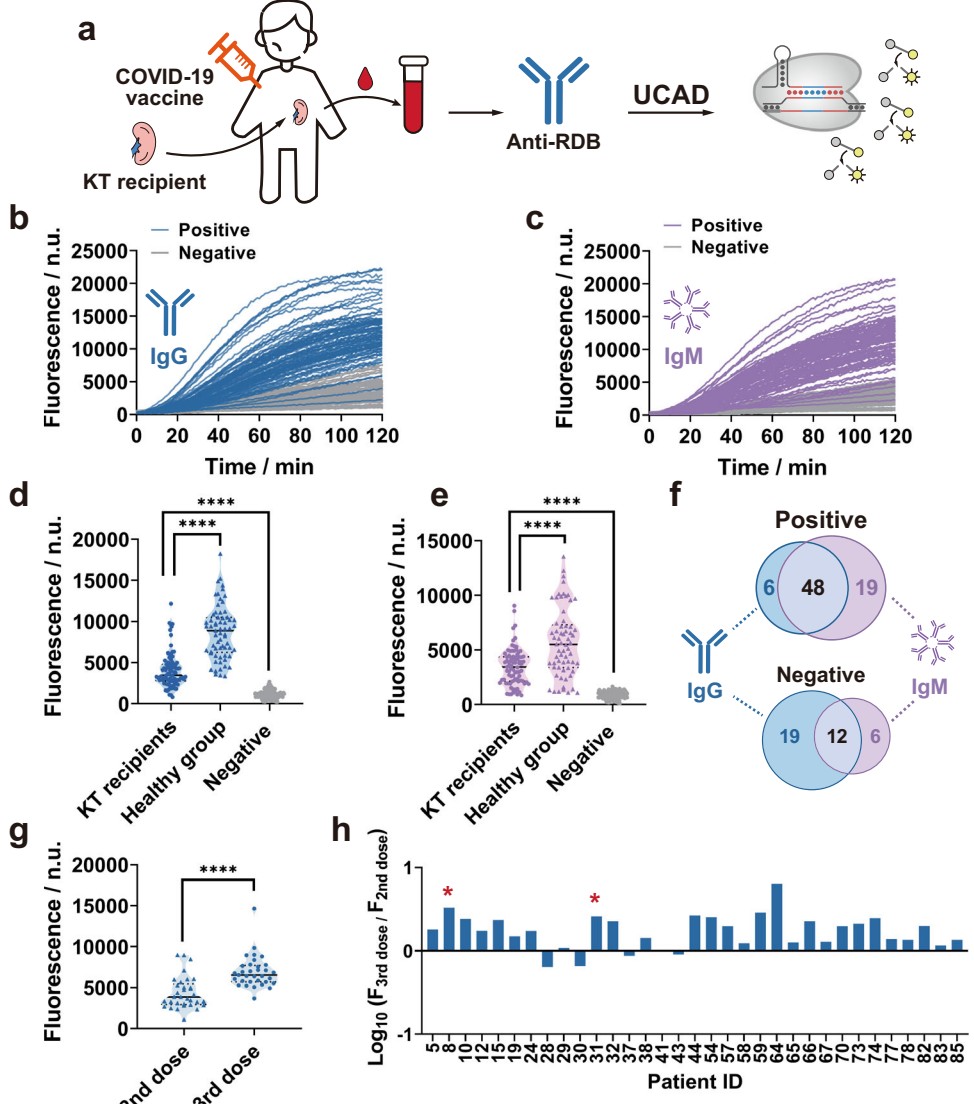

**Fig. 4 | Detection of anti-RBD IgG/IgM in vaccinated KTRs using UCAD.**
**a** Schematic illustration of the UCAD workflow for analyzing the levels of anti-RBD IgG and IgM in clinical serum samples collected from KTR patients who had received a triad immunosuppression regimen and two doses of inactivated COVID-19 vaccine. **b**, **c** Kinetic curves for the detection of anti-RBD IgG (**b**) and IgM (**c**) in sera of 85 KTRs. **d**, **e** Violin plots of endpoint fluorescence signals at 40 min for detecting anti-RBD IgG (**d**, $p = 1.76e^{-12}$ between KTRs and healthy, $p = 2.64e^{-19}$ between KTRs and negatives) and IgM (**e**, $p = 1.96e^{-7}$ between KTRs and healthy, $p = 4.18e^{-23}$ between KTRs and negatives) in 85 sera collected from KTRs, 65 positive sera collected from vaccinated healthy individuals, and 132 negative sera. The levels of both anti-RBD IgG and IgM in the KTR cohort were significantly different from those in the healthy and negative cohorts. Unpaired two-tailed $t$-tests were

performed to evaluate statistical differences between cohorts. **f** Venn diagrams to illustrate the distribution of positive and negative anti-RBD IgG and IgM test results in 85 KTRs. **g** Violin plots of endpoint fluorescence signals at 40 min for detecting serum anti-RBD IgG produced after the second and the third doses of inactivated COVID-19 vaccine in 33 KTRs. The anti-RBD IgG in KTRs received the third dose were significantly higher than their levels after the second dose of the vaccine. Unpaired two-tailed $t$-tests showed the statistic difference between the second dose and third dose cohorts was significant ($p = 2.22e^{-6}$). **h** Fold changes of anti-RBD IgG levels between the third and second vaccine doses in each KTR. Positive sera identified using standard CLIA IgG test were marked by *. Source data are available in the Source Data file. ****$p \leq 0.0001$.

The potential of UCAD for clinical uses was demonstrated by detecting anti-RBD IgG and IgM in 65 positive and 132 negative COVID-19 sera collected from the general population. By comparing to the standard CLIA test, we determined the sensitivity and specificity of UCAD to be 100 and 98.5%, respectively. The ultrahigh sensitivity of UCAD was further demonstrated in the detection of low levels of anti-SARS-CoV-2 IgG and IgM in a cohort of 85 KTRs who were defined to produce "undetectable" anti-SARS-CoV-2 antibodies using standard CLIA test. Because of therapeutic immunosuppression that impairs their immune responses to the COVID-19 vaccine, KTRs are an especially vulnerable patient population, with only 4–48% of KTRs having detectable anti-spike IgGs after receiving two vaccine doses[19,31–34].

However, it remains difficult to fully understand the humoral immunity of KTRs in response to COVID-19 infection or vaccination or to estimate the effectiveness of protective strategies such as the use of the third booster dose[19]. Here, by reclassifying anti-RBD-positive and anti-RBD-negative KTRs using UCAD, we found significant differences in the levels of plasmablasts, Th2 cells, and Th17 cells. Moreover, we found significant increases in anti-RBD IgG levels in 84.8% KTRs ($n = 33$) after receiving the third booster vaccine dose, confirming that the intensified vaccination approach is effective in KTRs. The fact that only two KTRs showed detectable IgG levels after the third vaccine dose by the standard CLIA suggests that more booster or higher antigen doses may be required to ensure sufficient protection in KTRs.

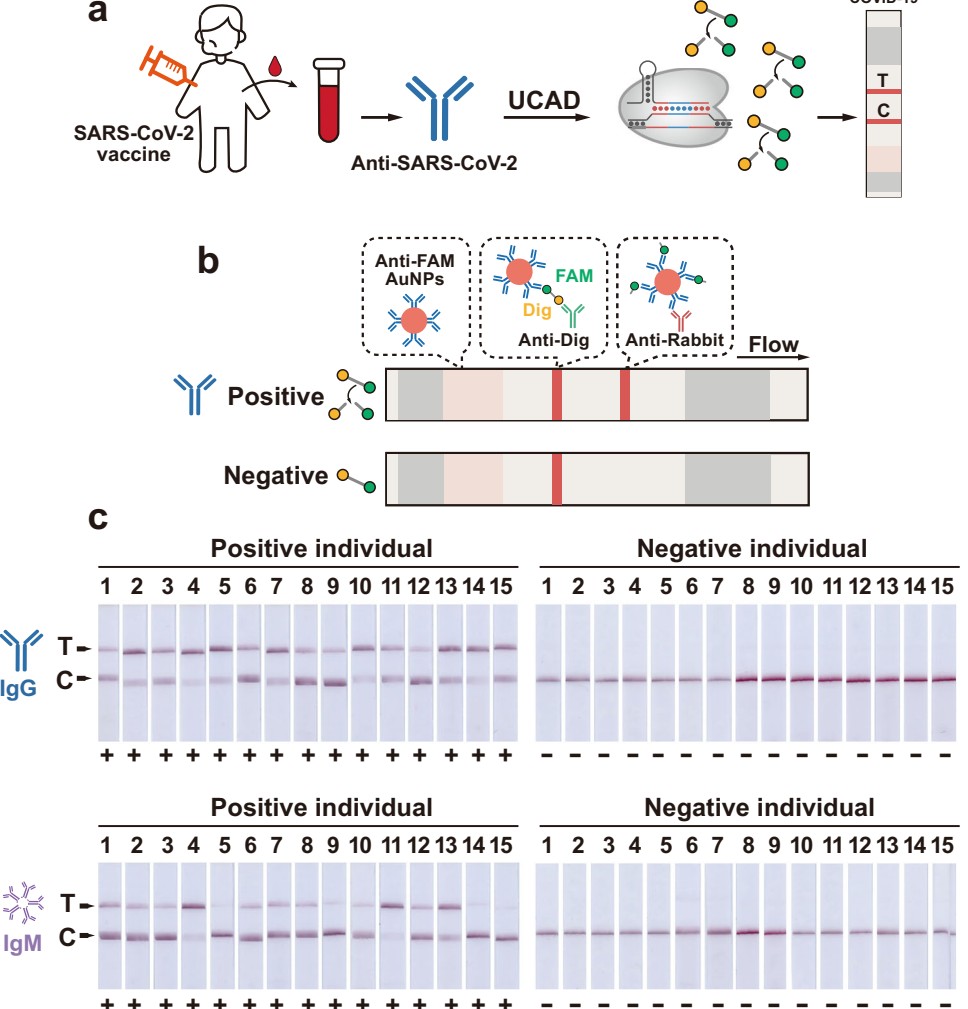

**Fig. 5 | Integrating UCAD with lateral flow readout. a** Schematic illustration of lateral flow readout for UCAD for the detection of anti-SARS-CoV-2 IgG and IgM. **b** The design of the lateral flow strip that captured all anti-FAM labeled AuNPs at C-line in the absence of the target antibody. In the presence of ant-SARS-CoV-2 IgG or IgM, the FAM-Dig dual-labeled reporters were cleaved by Cas12a-crRNA so that AuNPs could escape the capture at C-line and accumulated at T-line through the immobilized secondary antibody. **c** Visual detection of 15 positive clinical sera and 15 pre-pandemic sera by UCAD integrated with lateral flow strips.

In summary, this work demonstrates the development and clinical uses of UCAD as an ultrasensitive protein assay capable of redefining "undetectable" anti-SARS-CoV-2 antibody levels in clinical sera. In future studies, we aim to use this method to track temporal changes in antibody levels in infected and vaccinated individuals to gain a better understanding of the immunity of healthy people and immunocompromised groups. We will also further engineer UCAD as a broadly applicable POCT for the ultrasensitive detection of clinically important proteins beyond anti-SARS-CoV-2.

## Methods

### Ethical statement
This study was approved by the Ethics Committee of West China Hospital, Sichuan University (NO: 2021-110). All research was performed in accordance with relevant guidelines and regulations. All participants have provided informed consent.

### Sample collection
All clinical serum samples ($n = 315$) were obtained and analyzed using a standard CLIA test (2019-nCoV Ab, Xiamen InnoDx Biotechnology CO. LTD) with a cutoff value of 1.0. The 65 CLIA confirmed positive sera were collected from the West China Hospital of Sichuan University with the record of 2 vaccine doses and CLIA

positive. The 55 CLIA confirmed negative sera were also collected during the pandemic with the record of no vaccination and CLIA negative. All 77 pre-pandemic sera were collected from the bio-sample bank of West China Hospital of Sichuan University with the sampling date before the outbreak of COVID-19. No further selection has been made to either positive or negative samples. Blood samples from 85 KTRs were collected following the standard procedure at West China Hospital. Sera were prepared using serum separator tubes and incubated at room temperature for 1 h. The clot was removed by centrifugation at 1000×*g* for 10 min. The supernatant was collected, aliquoted, and stored at −80 °C upon usage. Peripheral blood mononuclear cells (PBMCs) were separated from EDTA anticoagulated blood samples of KTRs through density gradient separation[35], and the WBCs were counted and differentiated by the Sysmex hematology XN modular system.

### Preparation of DNA probes for UCAD
All DNA probes and reporters were purchased from Sangon Biotech Co., Ltd. (Shanghai, China) and were purified by high-performance liquid chromatography (HPLC), except crRNA, which was purchased from Integrated DNA Technologies, Inc., Coralville, USA. Sequences and modifications of all DNA or RNA probes are listed in Supplementary Table 1.

A biotin-streptavidin conjugation protocol was used to prepare all protein-conjugated DNA probes for UCAD. To prepare the RBD-conjugated DNA probe, 25 µL of 2.5 µM biotinylated TS probe (5′-TTCCTCACCATGTCTGAGGTACTCCTTAAAGGTCGAGCTGGAC-3′) was mixed with an equal volume of 2.5 µM streptavidin and incubated at 37 °C for 30 min. After cooling to room temperature, this reaction mixture was incubated with 50 µL of 1.25 µM biotinylated wild-type RBD (Sino Biological, Beijing, China), Delta RBD (L452R, T478K), or Omicron RBD (B.1.1.529) (ACROBiosystems, Beijing, China) of SARS-CoV-2 spike protein at 25 °C for another 30 min. After conjugation, the RBD-conjugated DNA probes were diluted to 250 nM with Tris-biotin buffer (20 mM Tris-HCl, 0.01% BSA, 1 mM biotin) and stored at 4 °C until usage. Biotinylated anti-human IgG or IgM (Sangon Biotech, Shanghai, China) conjugated NTS DNA probes (5′-TTGTTGAGGTAACCAAC-TATTTGTTACTGTTGCTTGTGGCCGTTTACGTCGCCGTCCAG-3′) were prepared using the same conjugation and storage protocols.

## UCAD protocol for the detection of anti-RBD in buffer, certified human sera, and clinical serum samples

For a typical UCAD assay, a 50 µL reaction mixture containing 5 µL serum sample (or buffer), 10 pM RBD-TS probe, 10 pM anti-human IgG (or IgM)-NTS probe, 40 µM dNTPs, and 2 units of T4 polymerase in 1 × NEBuffer™ 2 buffer was incubated at 37 °C for 20 min. For this reaction mixture, 10 µL solution was taken and mixed with reagents provided in the TwistDx™ RPA basic kit (TwistDx, Cambridge, UK) and incubated at 37 °C for 20 min to perform RPA. The RPA amplicon was then mixed with 40 nM Cas12a (Alt-R® A. s. Cas12a (Cpf1) V3, Integrated DNA Technologies, Inc., Coralville, USA), 40 nM crRNA (5′-UUAUUUCUACU CUUGUAGAUCGUCGCCGUCCAGCUCGACC-3′, Integrated DNA Technologies, Inc., Coralville, USA), and 40 nM FAM-BHQ-1 labeled ssDNA reporter in 100 µL 1× NEBuffer™ 2 in a 96-well microplate. Fluorescence was measured immediately with a data acquisition rate of one read per minute for 2 h at 37 °C using a Cytation 5 cell imaging multimode microplate reader (BioTek) with excitation/emission at 495/520 nm.

For lateral flow strip-based visual readout, the RPA amplicons were incubated with 40 nM Cas12a, 40 nM crRNA, and 40 nM FAM-Dig ssDNA reporter in 20 µL 1× NEBuffer™ 2 at 37 °C for 30 min and then loaded on the sample pad of the lateral flow strip with 30 µL of 4×SSC buffer. Once developed, the results on the lateral flow strips were read directly using the naked eye or captured using a digital camera.

## Detection of anti-RBD using standard ELISA test

A commercial ELISA kit (BGI genomics, Shenzhen, China) was used to measure anti-RBD IgG and IgM in certified COVID-19 positive or negative human serum samples included in the ELISA kit. The assay was performed according to the instructions provided by the manufacturer. Briefly, a 10 µL human serum sample (or diluted serum sample) was mixed with 100 µL diluent and incubated in the ELISA microplate well at 37 °C for 30 min. The microplate was then washed five times with 350 µL washing buffer. Then, 100 µL IgG or IgM enzymatic working solution was added to the microplate and incubated at 37 °C for 30 min. The microplate was washed again with 5 × 350 µL washing buffer. To this microplate, 50 µL substrate A and 50 µL substrate B were added and incubated at 37 °C for 10 min in darkness before the addition of 50 µL quenching solution. The OD value (450−620 nm) of each sample was measured immediately after quenching the reaction using Cytation 5 multimode microplate reader.

## Fabricate lateral flow strips for UCAD

The lateral flow strip was assembled by four components: the sample pad, the conjugate pad, the nitrocellulose membrane (Whatman®, purchased from Sigma Aldrich), and the absorbent pad. The sample pad was saturated with a buffer containing 0.25% Triton X-100, 0.05 M Tris-HCl, and 0.15 M NaCl. The optimized volume of AuNPs-anti-FITC conjugates were loaded on the conjugate pad. Digoxin rabbit polyclonal antibody was dispensed on the nitrocellulose membrane at the control (C) line, and the goat anti-rabbit IgG was dispensed at the test (T) line using the XYZ platform dispenser HM3030 (Shanghai Kinbio Tech Co., Ltd.). All the membranes and pads were dried at 37 °C for 2 h before being assembled and cut.

## Flow cytometric analysis of white blood cells

Seven IgG⁺/IgM⁺ KTRs, five IgG⁻/IgM⁻ KTRs, and five vaccinated healthy participants were recruited for the subpopulation analysis of WBCs by Fluorescence Activated Cell Sorter (FACS). About 2 ml EDTA anticoagulated clinical blood samples from each patient were collected. After lysing red blood cells as instructed[35], the WBC subsets were characterized by a BD FACSCanto™ II Cytometer after fluorescently labeling the surface antigens[36]. Thirty-four epitopes on the WBC surface were labeled using fluorophore-tagged antibodies, such as FITC, PE, PerCP-CY5.5, PC7, APC, APC-H7, PacificBlue, and BV-421. The 34 epitopes were CD45, CD3, CD4, CD8, CD25, CCR7, CD45RA, HLA-DR, CD19, CD56, CD127, IgG1, IgD, CD57, PD-1, CD28, CD27, CD38, CD86, CD123, 7-AAD, CD11c, Lin, CTLA-4, CCR10, CCR6, CCR4, CXCR3, CD160, LAG-3, Tim-3 CD15, CD33, CD11b, and CD14. Then labeled cells were sorted by a BD FACSCanto™ II Cytometer with antibody panels limited to eight colors. Data were analyzed by BD FACSDiva™ with a standardized gating strategy following the ONE-Study protocol (Supplementary Table 3)[37,38].

## Antibodies

SARS-CoV-2 Spike RBD human monoclonal antibody (Cat #: TA190325, clone name: OTIH401) and SARS-CoV-2 N protein human monoclonal antibody (Cat #: TA190323, clone name: OTIH1G5) were purchased from Origene, Beijing, China. SARS-CoV-2 Spike RBD rabbit polyclonal antibody (Cat #: A20135) was purchased from ABclonal Biotech, Wuhan, China. Mers-CoV Spike RBD human monoclonal antibody (Cat #: Q3731380, clone name: M336) was obtained from EMD Millipore, Darmstadt, Germany. SARS-CoV-2 Spike RBD human monoclonal antibody (Delta specific, Cat #: SPD-M370, clone name: AM110) and SARS-CoV-2 Spike RBD human monoclonal antibody (Omicron specific, Cat #: SPD-M415, clone name: AS113) were obtained from ACRObiosystems, Beijing, China. Digoxin rabbit monoclonal antibody (Cat #; A20267) was purchased from ABclonal Biotech, Wuhan, China. Biotin-conjugated goat anti-human IgG (Cat #: D110152-0100), biotin-conjugated goat anti-human IgM (Cat #: D110159-0100), and anti-FITC rabbit polyclonal antibody (Cat #: D110003-0200) were purchased from Sangon Biotech., Shanghai, China.

All antibodies were validated as described on the websites of their suppliers.

## Data analysis

The raw fluorescence data were analyzed by Microsoft 365 Excel and GraphPad 8.0.1. Thermodynamic data and melting temperatures were estimated using OligoAnalyzer (IDT), which is a web-based DNA calculator provided by Integrated DNA Technologies, Inc., and can be freely accessed at https://sg.idtdna.com/calc/analyzer. Flow cytometry data were analyzed by BD FACSDiva™ v9.0. To ensure assay accuracy and precision, a positive control containing a 10 nM dsDNA target and negative control of NEBuffer™ 2 buffer was included in all tests. All fluorescence signals were then normalized against the positive and negative controls to ensure data consistency.

## Statistics and reproducibility

No statistical method was used to predetermine sample size. No data were excluded from the analyses. The experiments were not randomized. The investigators were not blinded to allocation during experiments and outcome assessment. The unpaired two-tailed t-test was performed in Prism to compare UCAD signals of anti-RBD IgG and IgM between two cohorts. For comparisons between three or more

cohorts, ordinary one-way ANOVA was performed. Receiver operating characteristics (ROC) curve analysis was used in Prism to determine the cutoff values of UCAD for positive and negative results. A confusion matrix was performed using Excel to evaluate the performance of UCAD in comparison with the standard CLIA. 95% confidence intervals of UCAD sensitivity and specificity were calculated using the efficient-score method described by R. Newcombe[39]. For all analyses, a two-tailed $p$ value <0.05 was considered to be statistically significant.

### Reporting summary
Further information on research design is available in the Nature Research Reporting Summary linked to this article.

## Data availability
All data were available in the main text or the supplementary information text and files. The raw fluorescence data for all UCAD tests, unprocessed gel images, and data for flow cytometric analyses are provided in the Source data file. Source data are provided with this paper.

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

## Acknowledgements

We thank the Fundamental Research Funds for the Central Universities (No. YJ201975, F.L.), the National Natural Science Foundation of China (22006104, Y.T. and 22074099, F.L.), the Institutional Research Fund from Sichuan University (2021SCUNL105, F.L.), and the Sichuan Science and Technology Project (2021YJ0322, Y.T.) for financial supports.

## Author contributions

F.L. and T.L. conceived the idea, designed all experiments, and supervised the overall project. Y.T. and T.S. designed the UCAD assay protocol, performed all UCAD tests for all clinical serum samples, and completed data analysis. Y.T. and L.G. developed and optimized probes for the UCAD assay and performed ELISA tests. Y.T. designed and performed lateral flow tests. Y.T., T.S., and L.W. designed all clinical studies and performed clinical analysis. S.Y., M.M., and L.W. prepared all clinical serum samples and performed the flow cytometric analysis. Y.Y. and Y.W. performed the clinical validation and comparison between SARS-CoV-2 and SARS-CoV. All authors participated in the drafting and revision of the manuscript.

## Competing interests

Yanan Tang and Feng Li are inventors on a pending patent for UCAD technology by Sichuan University (Patent Number: PCT/CN2022/100758). The remaining authors declare no competing interests.
