## [Peer Review File · Nature Communications]

REVIEWER COMMENTS

Reviewer #1 (Remarks to the Author):

This is a technical description of a novel highly sensitive antibody detection assay with a brief application to a cohort of immunocompromised individuals. Authors have developed a CRISPR-based ultra-sensitive assay (UCAD) to detect anti-SARS-Cov2 RBD binding antibodies using a CRISPR-Cas12a pre-designed dsDNA barcode. They first describe the technical aspects and the sensitivity and specificity analyses, shortly describe a Point of care assay derived from their method, then apply the assay on a cohort of severely immunocompromised kidney transplant recipients who are non- or weak responders to the inactivated vaccine.

Though promising for future analysis and understanding of low antibody responses to SARS-Cov2 vaccines or other immunisations, a number of major concerns preclude any possible acceptance of the manuscript:

- the method is not enough described : while the kinetics of analysis is broadly shown it is not clearly stated at which time point should be performed the clinically adapted assays: a time point of 40mn is vaguely indicated but should be specified. This is a key point as the specificity tends to decline with time as shown by the kinetics curves for both IgG and IgM detection
- the specificity of the assay is insufficiently analyzed and does not appear to be strong enough:
 - it is checked first only between anti-RBD human and rabbit monoclonal Abs and anti-N Abs but should be also evaluated against other coronaviridae, (mainly SARS-CoV1) and various SARS-CoV2 VOCs and
 - then the clinical specificity analysis which mixes a combination of only 17 pre-pandemic sera with 55 post-pandemic ELISA negative sera as negative controls, shows a 3% rate of false positive results. This cannot be simply considered as outliers as authors did. They should precise whether these false positives were in the pre-pandemic or the CLIA-confirmed negative samples. Indeed the latter might have been exposed to viruses and have true low levels of Abs; it is requested the pre-pandemic cohort be expanded and that the specificity against various coronaviridae be analyzed in parallel to explain these false negative
- the POC assay is also weakly analyzed for its specificity in only 7 pre-pandemic sera.

If authors want to be convincing it is mandatory this specificity analysis is strengthened

Then authors apply their assay to a cohort of 85 KTR analyzed within large time ranges after 2 inactivated vaccine injections: 10 to 66 days. They show that while only 5 were positive with conventional ELISA, 73 become positive either for IgM and IgG with their UCAD, then show various correlations between these IgM or IGG responses with the patients clinical and immune characteristics. It would have been much more relevant to have a kinetics IgM and IgG analysis after the 1st and 2nd vaccine doses. Finally a missign information in this field is to evaluate whether this UCAD might help

discriminate whom of these UCAD-positive but ELISA-negative KTRs would benefit or not from a 3rd vaccine injection of the primary series or would rather be protected by passive administration of specific human monoclonal antibodies.

Reviewer #2 (Remarks to the Author):

This is not a unique study in a sense that it looked at serology but unique as I had not seen CRISPR in this field. The main issue is with no correlate of immune protection (COP) established, basing the premise on antibody and increasing sensitivity doesn't solve anything. We don't know what level of neutralizing Abs is even protective and even numerous reports showing re-infection or breakthrough infections in individuals with high antibody levels before the infection. So increasing the sensitivity like this is not the Holy Grail here.

Another issue why IgM is being measured here? Even if we assume neutralizing Ab is the COP.

Another main issue is low specificity. In a population with low seroprevalence, that would give us a very low positive predictive value if this is to be used in other populations with low vaccination coverage.

This is further useless when we are assessing the so called immunity against variants such as Omicron. Their RBD is apparently from the ancestral virus and not applicable to other variants.

All in all, despite neat basic science applied here, I don't envision much benefit for clinical applications. Using CRISPR for things like early HIV diagnostics would be enticing.

Reviewer #3 (Remarks to the Author):

The manuscript presented a very exciting research advancement that expands CRISPR diagnostics for the ultrasensitive detection of serum antibodies against SARS-CoV-2 in clinical settings through a quite unique design. This assay, termed UCAD, has achieved very impressive analytical performance for the detection of anti-RBD with LOD at attomolar level. More importantly, UCAD has also addressed an on-going clinical challenge for monitoring humoral immunity and the rate of seroconversion in kidney transplant patients, a very special group of immunocompromised population upon COVID-19 vaccination. Remarkably, the ultrasensitivity of UCAD allows the detection of serum IgG and IgM previously defined to be undetectable by conventional techniques and thus allows the reclassifying of

patients in response to COVID-19 vaccination. Given the novelty and technique advance of UCAD and strong clinical impact to COVID-19 pandemic, I would strongly recommend the acceptance of the work with some minor suggestions as follows.

1. CRISPR based protein detection has been previously established by several research groups. The authors shall cite these works and comment on how the current work advance existing CRISPR techniques for protein detection.
2. Despite UCAD shows acceptable quantification capacity, CRISPR diagnostic assays relying on RPA were generally semi-quantitative. The authors may discuss how to achieve better quantification capacity at the discussion section, for example, the use of microfluidic devices to achieve digital detection.
3. It's very nice to demonstrate that UCAD has the capability to be a point-of-care test by integrating with lateral flow readout. It's recommended to move this result from supplementary information to the main content.
4. Further to comment 3, UCAD is also compatible with other forms of readout for POC testing. The authors may expand this point in the discussion section.
5. As a diagnostic test for COVID-19, sometimes, it's necessary to detect antibodies against multiple antigens. Although this is not quite relevant to the current work, it will be nice to show UCAD can be modified to target antibodies against other antigens, such as the nucleocapsid protein of SARS-CoV-2.

Reviewer #4 (Remarks to the Author):

The authors developed a highly sensitive assay detecting anti-RBD antibodies. They first evaluated this assay in 65 anti-RBD positive and 72 anti-RBD negative sera collected from the general population, which showed high sensitivity and specificity. Next, they tested it on the vaccinated kidney transplant recipients, which showed seroconversion that were previously found to be undetectable.

The assay is very sensitive. However, the clinical utility is very unlimited. If the anti-RBD is undetectable with currently available commercial assays, the amount of antibody would be unlikely to achieve protection. Therefore, it is not useful for assessing vaccine response for patients. However, this assay may be useful in determining acute infection. The highly sensitive antibody assay may allow earlier detection of seroconversion.

Comments:

- In the first cohort, the number of negative controls (ie. CLIA negative samples) was only 72. The number is too small to determine an accurate specificity.
- Need more detailed description of the serum samples. For the CLIA RBD positive serum specimens in the first cohort (65 healthy individuals who have received 2 doses of vaccines), were these all patients in

that study? If not, how were these selected? Similarly, for the CLIA RBD negative serum specimens in the first cohort, were these all patients with specimens collected? If not, how were these selected? Details are required to understand whether there are potential bias in selecting serum specimens for evaluation.

- For the first cohort of 65 anti-RBD and 72 anti-RBD negative population: 1) When were the serum collected? 2) What is their level of antibodies as determined by the conventional RBD antibody assay?
- Provide the quantitative results for the CLIA RBD results for the positive and negative samples. This will allow readers to understand whether there are potential bias in selecting clearly positive or clearly negative specimens.
- Line 201: Need to add the confidence interval for all reported sensitivities and specificities.
- Figure 2: Does Figure 2D refer to IgG and figure 2E refer to IgM?
- Figure 2g: Should report the sensitivity/specificity of IgM and IgG separately. (ie. IgG in one figure and IgM in another figure)
- Line 282-285: Thought there is a significant correlation, it's very weak (r only ~ 0.25). Therefore, this is probably not clinically relevant.

Reviewer #5 (Remarks to the Author):

I read with great interest the article by Tang et al, entitled "A CRISPR-based ultrasensitive assay redefines undetectable SARS-CoV-2 antibodies in kidney transplant recipients after COVID-19 vaccination".

In their MS, the authors translate the detection of serum antibodies against the receptor binding domain (RBD) of SARS-CoV-2 spike protein into CRISPR-based nucleic acid testing in a homogeneous solution and demonstrate that this new technology, named Ultrasensitive CRISPR-based Antibody Detection (UCAD), is 10000 times more sensitive than the commercial immunoassay.

The article summarizes a massive amount experiments (for which the authors should be commended) and demonstrates convincingly that UCAD represents a technical advance for ultrasensitive protein analysis. Furthermore, because it does not require specialized equipment or tedious operational and washing steps, it could find wide applications for clinical uses in both centralized laboratories and point-of-care settings.

Major issues

Although I don't have much criticism on the technical aspect of the MS, I am far less convinced by the authors' attempt to demonstrate the interest of UCAD in the context of COVID vaccination in kidney transplant recipients.

Accumulating data have demonstrated that the protection offered by vaccination against COVID-19 to kidney transplant recipients (KTR), as for the general population depends on the viral neutralization capacity of patients' serum. Viral neutralization requires high amounts of antibodies, far above the threshold of detection of the available classical immunoassays. In contrast to what claimed by the authors in the abstract and introduction sections, being able to detect marginal amount of anti-spike IgGs after vaccination (which would not provide viral neutralization capacity) is therefore unlikely to affect the clinical management of patients.

Furthermore, most of the data related to identification of clinical variables or biomarkers (including flow cytometry analyses of PBMC) associated with response to vaccine of KTR have already been published elsewhere on larger cohorts of patients and bring very little to the present story.

Minor issues

It is methodologically wrong to calculate the sensitivity, specificity, precision and accuracy of a clinical assay based on results from a case control study. As the authors well know these values are impacted by the prevalence of the "disease" in the population. Panels f and g (and corresponding text) must be removed from the MS.

Response Letter

We thank all reviewers for the insightful comments and suggestions. We have thoroughly revised our manuscript accordingly. A point-by-point response is provided as following.

Reviewer #1:

This is a technical description of a novel highly sensitive antibody detection assay with a brief application to a cohort of immunocompromised individuals. Authors have developed a CRISPR-based ultra-sensitive assay (UCAD) to detect anti-SARS-Cov2 RBD binding antibodies using a CRISPR-Cas12a predesigned dsDNA barcode. They first describe the technical aspects and the sensitivity and specificity analyses, shortly describe a Point of care assay derived from their method, then apply the assay on a cohort of severely immunocompromised kidney transplant recipients who are non- or weak responders to the inactivated vaccine.

Though promising for future analysis and understanding of low antibody responses to SARS-Cov2 vaccines or other immunisations, a number of major concerns preclude any possible acceptance of the manuscript:

Comment-1: the method is not enough described: while the kinetics of analysis is broadly shown it is not clearly stated at which time point should be performed the clinically adapted assays: a time point of 40 min is vaguely indicated but should be specified. This is a key point as the specificity tends to decline with time as shown by the kinetics curves for both IgG and IgM detection.

Response-1: We agree with the reviewer that an optimal time point exist which offers the best trade-off between assay sensitivity and specificity. To address this concern, detailed kinetic analysis was included in Figure S4d. The result revealed that 40 min was the optimal time point that offered the widest dynamic range for quantitative analysis.

Revision-1: We included Figure S4d and corresponding discussion on choosing the optimal time point of UCAD in the revised manuscript.

Revision in the manuscript:

“Further kinetic analysis suggested that 40 min was optimal for endpoint fluorescence detection, as it offered the best dynamic range for quantitative analysis (Fig. S4d).”

Revision in SI:

Figure S4 | (d) Calibration curves using end-point fluorescence signals of mAb ranged from 10 aM to 1 pM at different time points of UCAD reaction. The fluorescence

signal at 40 min showed the best dynamic range and was thus selected as the best time point for subsequent experiments and data processing.

Comment-2: the specificity of the assay is insufficiently analyzed and does not appear to be strong enough: it is checked first only between anti-RBD human and rabbit monoclonal Abs and anti-N Abs but should be also evaluated against other coronaviridae, (mainly SARS-CoV1) and various SARS-CoV2 VOCs

Response-2: We thank the reviewer for raising this critical concern. We have expanded the specificity test against anti-MERS-CoV spike protein and anti-SARS-CoV in clinical sera. We have also evaluated the modularity and specificity of UCAD for detecting antibodies raised by Delta and Omicron mutants. A new section focusing on the specificity and modularity of UCAD was included as well as additional results in Figure 2, Figure S7, Figure S8, and Figure S9.

Revision-2: We included a new section on “Specificity and modularity of UCAD” and Figure 2 in the manuscript and Figure S7-S9 in SI.

Revision in the manuscript:

“**Specificity and modularity of UCAD.** UCAD was also determined to be highly specific, evidenced by the low cross-reactivity to human monoclonal antibodies against MERS-CoV spike protein RBD and SARS-CoV-2 nucleocapsid (N) protein, as well as rabbit polyclonal antibodies against SARS-CoV-2 throughout concentration ranges from 10 nM to 1 pM (Fig. 2a and Fig. S7). The specificity of UCAD was further validated against three anti-SARS-CoV positive human serum samples (Fig. S8). Results in Fig. 2b and 2c revealed background level of fluorescence in SARS-CoV serum samples (ANOVA, $p = 0.0676$ for IgG, $p = 0.1940$ for IgM), confirming the low cross-reactivity of UCAD against anti-SARS-CoV in clinical serum samples.

UCAD is also highly modifiable, the specificity of which can be engineered by simply switching the antibody-recognition domain conjugated with the TS probe (Fig. 2d). To demonstrate the modularity, we first modified TS probes with RBDs from the wild-type (WT) SARS-CoV-2, Delta mutant, and Omicron mutant, respectively, and validated the UCAD assay against monoclonal antibodies produced by the three subtypes of SARS-CoV-2 (Fig. 2e and Fig. S9). Excitingly, highest fluorescence signals were observed for all three types of anti-RBD antibodies only when correct RBD domain was employed for the UCAD assay throughout a concentration range from 100 nM to 1 pM. Moderate cross-reactivities were observed in Fig. 2e and Fig. S9, likely because of the conservative domains remained at the mutated RBDs, which was further confirmed by the lowest level of cross-reactivity between WT and Omicron. It is also possible to engineer the UCAD for the specific detection of anti-SARS-CoV-2 N protein by switching the RBD motif into the N protein (Fig. 2d). Results in Fig. 2f and Fig. S9d confirmed the high sensitivity for the detection of anti-SARS-CoV-2 N protein and low cross-reactivity with anti-RBD antibodies.”

Figure 2. (a) Detection of the target anti-human RBD and closely related nontargeted anti-MERS-CoV spike protein RBD human mAb (clone: m336), anti-SARS-CoV-2 nucleocapsid (N) protein human mAb and anti-SARS-CoV-2 spike protein RBD rabbit pAb at a concentration of 10 fM using UCAD. (b, c) Evaluation of the specificity of UCAD for anti-RBD IgG (b) and IgM (c) against anti-SARS-CoV in clinical sera from three SARS patients in 2003. The UCAD signals of anti-SARS-CoV positive sera were significantly lower than the anti-SARS-CoV-2 positive serum (ANOVA, $p = 0.0004$ for IgG, $p = 0.0025$ for IgM). (d) Schematic illustration of the modularity of UCAD for varying mutants of RBD and the N protein of SARS-CoV-2 by switching the recognition motif on the TS probe. (e) Heatmap of the detection of 10 fM wild-type (WT), Delta specific and Omicron specific anti-RBD human mAb by using WT RBD, Delta RBD and Omicron RBD (B.1.1.529) modified TS probes. (f) Heatmap of the detection of 10 fM WT RBD human mAb and anti-N protein human mAb with WT RBD and N protein modified TS probes.

Revision in SI:

Figure S7 | Specificity of UCAD. Evaluating the specificity of UCAD for anti-SARS-CoV-2 spike protein RBD human mAb (SARS-CoV-2 RBD human) against closely related anti- MERS-CoV spike protein RBD human mAb (MER-CoV RBD human), anti-SARS-CoV-2 spike protein RBD rabbit pAb (SARS-CoV-2 RBD rabbit) and anti-SARS-CoV-2 N nucleocapsid protein mAb (SARS-CoV-2 N human). (a) UCAD kinetic curves for 10 fM anti-SARS-CoV-2 RBD against equal concentrations of nonspecific antibodies. (b) Detection of SARS-CoV-2 RBD human mAb and nonspecific antibodies in the concentration range from 1 pM to 10 aM using UCAD. Only SARS-CoV-2 RBD human mAb generated fluorescence signal linearly correlated

with its concentration. [TS probe] = 10 pM, [NTS probe] = 10 pM, [T4 polymerase] = 2 U, [dNTPs] = 40 μM, [Cas12a] = 40 nM, [crRNA] = 40 nM, [Reporter] = 40 nM.

Figure S8 | Validation of the SARS-CoV serum sample. A standard ELISA test was performed to validate the presence of anti-SARS-CoV IgG in three human serum samples from patients with confirmed SARS-CoV infections. Serum samples from six healthy individuals were also included to ensure the specificity of the ELISA test.

Figure S9 | Modularity of UCAD. (a) Detection of human anti-RBD mAbs against wild-type (WT) SARS-CoV-2, Delta mutant, and Omicron mutant using a WT-specific TS probe. (b) Detection of human anti-RBD mAbs against wild-type (WT) SARS-CoV-2, Delta mutant, and Omicron mutant using a Delta-specific TS probe. (c) Detection of human anti-RBD mAbs against wild-type (WT) SARS-CoV-2, Delta mutant, and Omicron mutant using an Omicron-specific TS probe. (d) Detection of anti-SARS-CoV-2 nucleocapsid (N) human mAb (anti-N protein) and anti-RBD using N protein-specific TS probe. All antibodies were tested in a concentration range from 1 pM to 100 aM. [TS probe] = 10 pM, [NTS probe] = 10 pM, [T4 polymerase] = 2 U, [dNTPs] = 40 μM, [Cas12a] = 40 nM, [crRNA] = 40 nM, [Reporter] = 40 nM.

Comment-3: the clinical specificity analysis which mixes a combination of only 17 pre-pandemic sera with 55 post-pandemic ELISA negative sera as negative controls, shows a 3% rate of false positive results. This cannot be simply considered as outliers as authors did. They should precise whether these false positives were in the pre-pandemic or the CLIA-confirmed negative samples. Indeed, the latter might have been exposed to viruses and have true low levels of Abs; it is requested the pre-pandemic cohort be expanded and that the specificity against various coronaviridae be analyzed in parallel to explain these false negative.

Response-3: As suggested by the reviewer, we have expanded the pre-pandemic cohort from n = 17 to n = 77. None of the pre-pandemic sample shows false positive. The only 2 weak false positives were in the 55 post-pandemic CLIA negative sera, so there is possibility that the two participants had exposed to viruses and produced low level of antibodies. Therefore, we removed the statement about the outliers and included more

specific discussion on the possible source for the false positives. The specificity test against SARS-CoV clinical samples were also included as outlined above in Response-2.

Revision-3: We have revised the manuscript and Figure 3 with expanded pre-pandemic cohort. We have also included the discussion of the possible source for the false positives.

“As the two weak false positives were both in the cohort of the 55 post-pandemic CLIA negative sera, there is possibility that the two participants had exposed to viruses and produced low level of anti-RBD below the LOD of CLIA.”

Comment-4: the POC assay is also weakly analyzed for its specificity in only 7 pre-pandemic sera. If authors want to be convincing it is mandatory this specificity analysis is strengthened.

Response-4: We have expanded the pre-pandemic sera to be equal to the positive sera (n = 15) in the revised manuscript.

Comment-5: Then authors apply their assay to a cohort of 85 KTR analyzed within large time ranges after 2 inactivated vaccine injections: 10 to 66 days. They show that while only 5 were positive with conventional ELISA, 73 become positive either for IgM and IgG with their UCAD, then show various correlations between these IgM or IgG responses with the patients clinical and immune characteristics. It would have been much more relevant to have a kinetics IgM and IgG analysis after the 1st and 2nd vaccine doses.

Response-5: We agree that it would be ideal to monitor the kinetic changes of serum IgG and IgM levels after the 1st and 2nd vaccine doses. However, this is practically not possible for us, because KTRs included in this work were highly scattered in the West China area and we could only collect their blood samples during their regular follow-ups. Instead, we monitored the changes of serum IgG levels between the 2nd and 3rd dose of COVID-19 vaccine for 33 KTRs and included this result in Figure 4g, 4h and Figure S33.

Revision-5: We have revised manuscript by including the results and discussion of changes of serum IgG levels between the 2nd and 3rd dose of COVID-19 vaccine.

Revision in the manuscript:

“We also employed UCAD to monitor the changes of anti-RBD IgG levels in 33 KTRs who received the 3rd dose of inactivated COVID-19 vaccine. Sera were collected between 8-179 days (median = 94) after receiving the 3rd dose of vaccine. All 33 KTRs were found to be anti-RBD IgG negative after the 2nd vaccine dose and only 2 were found to turn positive after the 3rd dose using the standard CLIA (Table S4). By quantifying the “undetectable” levels of anti-RBD IgG using UCAD, we observed significant elevation of anti-RBD IgG levels in 28 out of 33 (84.8%) KTRs after

receiving the 3rd dose with fold changes as much as 6.4 times (Fig. 4g, 4h and S33).”

Figure 4. (g) Violin plots of end-point fluorescence signals at 40 min for detecting serum anti-RBD IgG produced after the 2nd and the 3rd doses of inactivated COVID-19 vaccine in 33 KTRs. The anti-RBD IgG in KTRs received the 3rd dose were significantly higher than their levels after the 2nd dose of vaccine. ($p < 0.0001$) (h) Fold changes of anti-RBD IgG levels between the 3rd and 2nd vaccine doses in each KTR. Positive sera identified using standard CLIA IgG test were marked by *.

Revision in SI:

Figure S33 | UCAD analysis of KTRs with the 3rd dose of vaccine. (a) Kinetic curves for the detection of anti-RBD IgG in the 33 KTRs who received the 3rd dose of inactivated COVID-19 vaccine. (b) Comparison of anti-RBD IgG signals of the 33 KTRs received the 2nd and 3rd dose of vaccine.

Comment-6: Finally, a missing information in this field is to evaluate whether this UCAD might help discriminate whom of these UCAD-positive but ELISA-negative KTRs would benefit or not from a 3rd vaccine injection of the primary series or would rather be protected by passive administration of specific human monoclonal antibodies.

Response-6: We agree and included the study of the effect of the 3rd dose of vaccine in the revised manuscript as outlined in Response-5.

Reviewer #2:

This is not a unique study in a sense that it looked at serology but unique as I had not seen CRISPR in this field.

Comment-1: The main issue is with no correlate of immune protection (COP) established, basing the premise on antibody and increasing sensitivity doesn't solve anything. We don't know what level of neutralizing Abs is even protective and even numerous reports showing re-infection or breakthrough infections in individuals with

high antibody levels before the infection. So increasing the sensitivity like this is not the Holy Grail here.

Response-1: We agree with the reviewer that increasing the sensitivity for antibody test did not provide immediate solution for better immune protection. However, an ultrasensitive protein assay can be used for early diagnosis and for providing important information for understanding the fundamental basis of immune response to infection or vaccination. Therefore, we have reorganized and thoroughly revised manuscript to focus more on the technological aspect of UCAD rather than its immediate clinical impact to COVID-19. We discussed the design principle and analytical performance in terms of its sensitivity, specificity, and generalizability to different targets. We also discussed its potential applications to COVID-19 and other infectious or somatic diseases.

Comment-2: Another issue why IgM is being measured here? Even if we assume neutralizing Ab is the COP.

Response-2: We agree that the level of IgM might not be closely related to COP upon vaccination, but it could be an important marker for early diagnosis. As outlined in response-1, we have thoroughly revised manuscript to focus more on the technological aspect of UCAD rather than its immediate clinical impact to COVID-19.

Comment-3: Another main issue is low specificity. In a population with low seroprevalence, that would give us a very low positive predictive value if this is to be used in other populations with low vaccination coverage.

Response-3: We agree, and we could not answer the question how UCAD performs at different populations with current data and this is much beyond the scope of our current work. Therefore, we have thoroughly revised manuscript to focus more on the technological aspect of UCAD rather than its immediate clinical impact to COVID-19.

Comment-4: This is further useless when we are assessing the so called immunity against variants such as Omicron. Their RBD is apparently from the ancestral virus and not applicable to other variants.

Response-4: In the revised manuscript, we demonstrate that UCAD is highly modulable and can be engineered to respond to antibodies against the wild-type virus as well as Delta and Omicron mutants by switching the recognition motif to corresponding RBD.

Revision-4: We included a new section on “Specificity and modularity of UCAD” and Figure 2 in the manuscript and Figure S9 in SI to demonstrate the modularity of UCAD. Revision in the manuscript:

“UCAD is also highly modulable, the specificity of which can be engineered by simply

switching the antibody-recognition domain conjugated with the TS probe (Fig. 2d). To demonstrate the modularity, we first modified TS probes with RBDs from the wild-type (WT) SARS-CoV-2, Delta mutant, and Omicron mutant, respectively, and validated the UCAD assay against monoclonal antibodies produced by the three subtypes of SARS-CoV-2 (Fig. 2e and Fig. S9). Excitingly, highest fluorescence signals were observed for all three types of anti-RBD antibodies only when correct RBD domain was employed for the UCAD assay throughout a concentration range from 100 aM to 1 pM. Moderate cross-reactivities were observed in Fig. 2e and Fig. S9, likely because of the conservative domains remained at the mutated RBDs, which was further confirmed by the lowest level of cross-reactivity between WT and Omicron. It is also possible to engineer the UCAD for the specific detection of anti-SARS-CoV-2 N protein by switching the RBD motif into the N protein (Fig. 2d). Results in Fig. 2f and Fig. S9d confirmed the high sensitivity for the detection of anti-SARS-CoV-2 N protein and low cross-reactivity with anti-RBD antibodies.??

Figure 2. (a) Detection of the target anti-human RBD and closely related nontargeted anti-MERS-CoV spike protein RBD human mAb (clone: m336), anti-SARS-CoV-2 nucleocapsid (N) protein human mAb and anti-SARS-CoV-2 spike protein RBD rabbit pAb at a concentration of 10 fM using UCAD. (b, c) Evaluation of the specificity of UCAD for anti-RBD IgG (b) and IgM (c) against anti-SARS-CoV in clinical sera from three SARS patients in 2003. The UCAD signals of anti-SARS-CoV positive sera were significantly lower than the anti-SARS-CoV-2 positive serum (ANOVA, $p = 0.0004$ for IgG, $p = 0.0025$ for IgM). (d) Schematic illustration of the modularity of UCAD for varying mutants of RBD and the N protein of SARS-CoV-2 by switching the recognition motif on the TS probe. (e) Heatmap of the detection of 10 fM wild-type (WT), Delta specific and Omicron specific anti-RBD human mAb by using WT RBD, Delta RBD and Omicron RBD (B.1.1.529) modified TS probes. (f) Heatmap of the detection of 10 fM WT RBD human mAb and anti-N protein human mAb with WT RBD and N protein modified TS probes.

Revision in SI:

Figure S9 | Modularity of UCAD. (a) Detection of human anti-RBD mAbs against wild-type (WT) SARS-CoV-2, Delta mutant, and Omicron mutant using a WT-specific TS probe. (b) Detection of human anti-RBD mAbs against wild-type (WT) SARS-CoV-2, Delta mutant, and Omicron mutant using a Delta-specific TS probe. (c) Detection of human anti-RBD mAbs against wild-type (WT) SARS-CoV-2, Delta mutant, and Omicron mutant using an Omicron-specific TS probe. (d) Detection of anti-SARS-CoV-2 nucleocapsid (N) human mAb (anti-N protein) and anti-RBD using N protein-specific TS probe. All antibodies were tested in a concentration range from 1 pM to 100 aM. [TS probe] = 10 pM, [NTS probe] = 10 pM, [T4 polymerase] = 2 U, [dNTPs] = 40 μM, [Cas12a] = 40 nM, [crRNA] = 40 nM, [Reporter] = 40 nM.

Comment-5: All in all, despite neat basic science applied here, I don't envision much benefit for clinical applications. Using CRISPR for things like early HIV diagnostics would be enticing.

Response-5: As outlined in abovementioned responses, we have thoroughly revised manuscript to focus more on the technological aspect of UCAD and demonstrated that it indeed possible to engineer UCAD to detect other targets by changing the recognition motif on the TS probe.

Reviewer #3:

The manuscript presented a very exciting research advancement that expands CRISPR diagnostics for the ultrasensitive detection of serum antibodies against SARS-CoV-2 in clinical settings through a quite unique design. This assay, termed UCAD, has achieved very impressive analytical performance for the detection of anti-RBD with LOD at attomolar level. More importantly, UCAD has also addressed an on-going clinical challenge for monitoring humoral immunity and the rate of seroconversion in kidney transplant patients, a very special group of immunocompromised population upon COVID-19 vaccination. Remarkably, the ultrasensitivity of UCAD allows the detection of serum IgG and IgM previously defined to be undetectable by conventional techniques and thus allows the reclassifying of patients in response to COVID-19 vaccination. Given the novelty and technique advance of UCAD and strong clinical impact to COVID-19 pandemic, I would strongly recommend the acceptance of the work with some minor suggestions as follows.

Comment-1: CRISPR based protein detection has been previously established by several research groups. The authors shall cite these works and comment on how the current work advance existing CRISPR techniques for protein detection.

Response-1: We thank the reviewer for this suggestion. Indeed, there has been a couple works focusing on engineering CRISPR for protein detection, including a CRISPR-based immunosorbent assay developed by Zhou lab (Anal. Chem. 2020, 92, 573-577) and a proximity CRISPR assay previously introduced by our own lab (Chem. Sci. 2021, 12, 2133-2137). However, unlike UCAD, neither effort has achieved LOD comparable

with CRISPR-based nucleic acid testing. Another important advancement was that UCAD had been vigorously validated against clinical samples and all other CRISPR-based protein assays were only tested on model systems rather than clinical samples. We included this discussion in the revised manuscript and cited the corresponding literatures.

Revision-1: We included the discussion on the comparison of UCAD and other CRISPR-based protein detection as well as corresponding citations in the revised manuscript.

“Comparing existing efforts to harness CRISPR systems for protein detection, UCAD demonstrates remarkable higher sensitivity and lower LOD not only in buffer systems but also in real clinical settings.^{24,25}”

Comment-2: Despite UCAD shows acceptable quantification capacity, CRISPR diagnostic assays relying on RPA were generally semi-quantitative. The authors may discuss how to achieve better quantification capacity at the discussion section, for example, the use of microfluidic devices to achieve digital detection.

Response-2: We have revised the manuscript by adding the discussion on the potential to integrate UCAD with other detection platforms including the droplet microfluidic system to further enhance the sensitivity and quantification capacity.

Revision-2: The following discussion and citations were added to the revised manuscript.

“Because of the simple, homogeneous nature of UCAD, it is also possible to integrate UCAD with many other visual, colorimetric detection platform, such as the use of plasmonic gold nanoparticles,²⁶ for field-based applications or with droplet microfluidic systems²⁷ to further enhance the sensitivity and quantification capacity.”

Comment-3: It’s very nice to demonstrate that UCAD has the capability to be a point-of-care test by integrating with lateral flow readout. It’s recommended to move this result from supplementary information to the main content.

Response-3: We thank the reviewer for this suggestion and have moved this result in the revised manuscript as Figure 5.

Comment-4: Further to comment 3, UCAD is also compatible with other forms of readout for POC testing. The authors may expand this point in the discussion section.

Response-4: As outlined in Response-2, we have revised the manuscript by adding the discussion on the potential to integrate UCAD with other detection platforms including other forms of readout for POCT.

Comment-5: As a diagnostic test for COVID-19, sometimes, it’s necessary to detect

antibodies against multiple antigens. Although this is not quite relevant to the current work, it will be nice to show UCAD can be modified to target antibodies against other antigens, such as the nucleocapsid protein of SARS-CoV-2.

Response-5: We thank the reviewer for this suggestion and have demonstrated modularity of UCAD by engineering it to detect antibodies against SARS-CoV-2 N protein as well as RBD of Delta and Omicron variants. We have included a new “Specificity and Modularity” section and Figure 2 and S9 in the revised manuscript.

Revision-5: We included a new section on “Specificity and modularity of UCAD” and Figure 2 in the manuscript and Figure S9 in SI to demonstrate the modularity of UCAD. Revision in the manuscript:

“UCAD is also highly modifiable, the specificity of which can be engineered by simply switching the antibody-recognition domain conjugated with the TS probe (Fig. 2d). To demonstrate the modularity, we first modified TS probes with RBDs from the wild-type (WT) SARS-CoV-2, Delta mutant, and Omicron mutant, respectively, and validated the UCAD assay against monoclonal antibodies produced by the three subtypes of SARS-CoV-2 (Fig. 2e and Fig. S9). Excitingly, highest fluorescence signals were observed for all three types of anti-RBD antibodies only when correct RBD domain was employed for the UCAD assay throughout a concentration range from 100 nM to 1 pM. Moderate cross-reactivities were observed in Fig. 2e and Fig. S9, likely because of the conservative domains remained at the mutated RBDs, which was further confirmed by the lowest level of cross-reactivity between WT and Omicron. It is also possible to engineer the UCAD for the specific detection of anti-SARS-CoV-2 N protein by switching the RBD motif into the N protein (Fig. 2d). Results in Fig. 2f and Fig. S9d confirmed the high sensitivity for the detection of anti-SARS-CoV-2 N protein and low cross-reactivity with anti-RBD antibodies.”

Figure 2. (a) Detection of the target anti-human RBD and closely related nontargeted anti-MERS-CoV spike protein RBD human mAb (clone: m336), anti-SARS-CoV-2 nucleocapsid (N) protein human mAb and anti-SARS-CoV-2 spike protein RBD rabbit pAb at a concentration of 10 fM using UCAD. (b, c) Evaluation of the specificity of UCAD for anti-RBD IgG (b) and IgM (c) against anti-SARS-CoV in clinical sera from three SARS patients in 2003. The UCAD signals of anti-SARS-CoV positive sera were significantly lower than the anti-SARS-CoV-2 positive serum (ANOVA, $p = 0.0004$ for IgG, $p = 0.0025$ for IgM). (d) Schematic illustration of the modularity of UCAD for

varying mutants of RBD and the N protein of SARS-CoV-2 by switching the recognition motif on the TS probe. (e) Heatmap of the detection of 10 fM wild-type (WT), Delta specific and Omicron specific anti-RBD human mAb by using WT RBD, Delta RBD and Omicron RBD (B.1.1.529) modified TS probes. (f) Heatmap of the detection of 10 fM WT RBD human mAb and anti-N protein human mAb with WT RBD and N protein modified TS probes.

Revision in SI:

Figure S9 | Modularity of UCAD. (a) Detection of human anti-RBD mAbs against wild-type (WT) SARS-CoV-2, Delta mutant, and Omicron mutant using a WT-specific TS probe. (b) Detection of human anti-RBD mAbs against wild-type (WT) SARS-CoV-2, Delta mutant, and Omicron mutant using a Delta-specific TS probe. (c) Detection of human anti-RBD mAbs against wild-type (WT) SARS-CoV-2, Delta mutant, and Omicron mutant using an Omicron-specific TS probe. (d) Detection of anti-SARS-CoV-2 nucleocapsid (N) human mAb (anti-N protein) and anti-RBD using N protein-specific TS probe. All antibodies were tested in a concentration range from 1 pM to 100 aM. [TS probe] = 10 pM, [NTS probe] = 10 pM, [T4 polymerase] = 2 U, [dNTPs] = 40 μM, [Cas12a] = 40 nM, [crRNA] = 40 nM, [Reporter] = 40 nM.

Reviewer #4:

The authors developed a highly sensitive assay detecting anti-RBD antibodies. They first evaluated this assay in 65 anti-RBD positive and 72 anti-RBD negative sera collected from the general population, which showed high sensitivity and specificity. Next, they tested it on the vaccinated kidney transplant recipients, which showed seroconversion that were previously found to be undetectable.

The assay is very sensitive. However, the clinical utility is very unlimited. If the anti-RBD is undetectable with currently available commercial assays, the amount of antibody would be unlikely to achieve protection. Therefore, it is not useful for assessing vaccine response for patients. However, this assay may be useful in determining acute infection. The highly sensitive antibody assay may allow earlier detection of seroconversion.

Comment-1: In the first cohort, the number of negative controls (ie. CLIA negative samples) was only 72. The number is too small to determine an accurate specificity.

Response-1: We thank the reviewer for raising this concern. We agree that CLIA negative samples, in particular, the pre-pandemic negatives needed to be expanded, therefore, we have expanded the pre-pandemic cohort from n = 17 to n = 77 and the total negatives became n = 132 in the revised manuscript.

Comment-2: Need more detailed description of the serum samples. For the CLIA RBD

positive serum specimens in the first cohort (65 healthy individuals who have received 2 doses of vaccines), were these all patients in that study? If not, how were these selected? Similarly, for the CLIA RBD negative serum specimens in the first cohort, were these all patients with specimens collected? If not, how were these selected? Details are required to understand whether there are potential bias in selecting serum specimens for evaluation.

Response-2: The 65 CLIA positive sera were collected from the West China Hospital of Sichuan University with the record of 2 vaccine doses and CLIA positive. The 55 CLIA negative sera were also collected during pandemic with the record of no vaccination and CLIA negative. All 77 pre-pandemic sera were collected from the sample bank of West China Hospital of Sichuan University with the sampling date before the outbreak of COVID-19. No further selection has been made to either positive or negative samples. Details of sampling information were added in the revised manuscript.

Revision-2: We have added the information in the Sample Collection section of the revised manuscript.

“The 65 CLIA positive confirmed sera were collected from the West China Hospital of Sichuan University with the record of 2 vaccine doses and CLIA positive. The 55 CLIA confirmed negative sera were also collected during pandemic with the record of no vaccination and CLIA negative. All 77 pre-pandemic sera were collected from the bio-sample bank of West China Hospital of Sichuan University with the sampling date before the outbreak of COVID-19. No further selection has been made to either positive or negative samples.”

Comment-3: For the first cohort of 65 anti-RBD and 72 anti-RBD negative population: 1) When were the serum collected? 2) What is their level of antibodies as determined by the conventional RBD antibody assay? Provide the quantitative results for the CLIA RBD results for the positive and negative samples. This will allow readers to understand whether there are potential bias in selecting clearly positive or clearly negative specimens.

Response-3: We thank the reviewer for this critical suggestion and have added the demographic information and CLIA test results of the 65 positive and 55 in-pandemic negative participants in Table S2 and S3 in the revised manuscript.

Comment-4: Line 201: Need to add the confidence interval for all reported sensitivities and specificities.

Response-4: We calculated the confidence intervals for the sensitivity and specificity using the efficient-score method described by R. Newcombe (ref39) and added this information in the revised manuscript.

Revision-4: The confidence intervals were added in the caption of Figure 3 of the revised manuscript.

“The sensitivity of UCAD was 100% (95% confidence interval: 93.0% - 100%) and its specificity was 98.5% (95% confidence interval: 94.1% - 99.7%).”

Comment-5: Figure 2: Does Figure 2D refer to IgG and figure 2E refer to IgM?

Response-5: Yes, we have added the labels of IgG and IgM in the revised manuscript.

Comment-6: Figure 2g: Should report the sensitivity/specificity of IgM and IgG separately. (ie. IgG in one figure and IgM in another figure)

Response-6: We thank the reviewer for the suggestion. We agree that it would be ideal to report the sensitivity and specificity of IgM and IgG separately. However, only total anti-RBD test was performed for healthy participants at the West China Hospital of Sichuan University. Therefore, we pooled the binary results of UCAD IgG and IgM to determine the sensitivity and specificity of UCAD.

Comment-7: Line 282-285: Thought there is a significant correlation, it's very weak (r only ~0.25). Therefore, this is probably not clinically relevant.

Response-7: We agree this is probably not clinically relevant. As suggested by the editor, we have reorganized and thoroughly revised manuscript to focus more on the technological aspect of UCAD and this component was removed in the revised manuscript.

Reviewer #5:

I read with great interest the article by Tang et al, entitled “A CRISPR-based ultrasensitive assay redefines undetectable SARS-CoV-2 antibodies in kidney transplant recipients after COVID-19 vaccination”. In their MS, the authors translate the detection of serum antibodies against the receptor binding domain (RBD) of SARS-CoV-2 spike protein into CRISPR-based nucleic acid testing in a homogeneous solution and demonstrate that this new technology, named Ultrasensitive CRISPR-based Antibody Detection (UCAD), is 10000 times more sensitive than the commercial immunoassay. The article summarizes a massive amount of experiments (for which the authors should be commended) and demonstrates convincingly that UCAD represents a technical advance for ultrasensitive protein analysis. Furthermore, because it does not require specialized equipment or tedious operational and washing steps, it could find wide applications for clinical uses in both centralized laboratories and point-of-care settings.

Major issues

Comment-1: Although I don't have much criticism on the technical aspect of the MS, I am far less convinced by the authors' attempt to demonstrate the interest of UCAD in

the context of COVID vaccination in kidney transplant recipients. Accumulating data have demonstrated that the protection offered by vaccination against COVID-19 to kidney transplant recipients (KTR), as for the general population depends on the viral neutralization capacity of patients' serum. Viral neutralization requires high amounts of antibodies, far above the threshold of detection of the available classical immunoassays. In contrast to what claimed by the authors in the abstract and introduction sections, being able to detect marginal amount of anti-spike IgGs after vaccination (which would not provide viral neutralization capacity) is therefore unlikely to affect the clinical management of patients.

Response-1: We agree with the reviewer that UCAD may not be able to generate immediate impact to the clinical management of KTRs. It is better suited for other clinical applications, such as early diagnosis or detecting protein-based biomarkers for infection or other diseases. Therefore, we have reorganized and thoroughly revised manuscript to shift the emphasis from clinical utility to broader technical advancement and clinical validation of UCAD technology for potential applications, such as early diagnosis, point-of-care testing, as well as monitoring the changes in antibody levels upon vaccination and the use of the booster dose in immunocompromised population.

Comment-2: Furthermore, most of the data related to identification of clinical variables or biomarkers (including flow cytometry analyses of PBMC) associated with response to vaccine of KTR have already been published elsewhere on larger cohorts of patients and bring very little to the present story.

Response-2: We agree. As outlined in Response-1, we have reorganized and thoroughly revised manuscript to shift the emphasis from clinical utility to broader technical advancement and clinical validation of UCAD technology. Sera antibody tests of KTRs were still included to validate the ultrahigh sensitivity of UCAD in clinical settings. We also added new test results of the anti-RBD IgG level of KTRs after receiving the 3rd dose of vaccine to further verify the possibility to use UCAD to monitor changes of antibody levels in a detection window not achievable by conventional techniques, such as CLIA.

Minor issues

Comment-3: It is methodologically wrong to calculate the sensitivity, specificity, precision and accuracy of a clinical assay based on results from a case control study. As the authors well know these values are impacted by the prevalence of the "disease" in the population. Panels f and g (and corresponding text) must be removed from the MS.

Response-3: We agree with the reviewer that it is incorrect to calculate the precision and accuracy of UCAD. We removed this component from Figure 3g and corresponding discussion. However, it remains possible to determine the sensitivity and specificity of UCAD by comparing to the standard CLIA test, which were retained in the revised manuscript.

REVIEWERS' COMMENTS

Reviewer #1 (Remarks to the Author):

Authors have answered adequately the major concerns raised by the first version and this revised manuscript has been strongly improved. The addition of more specificity investigations such as against the SARS-COV2 variants is promising although the UCAD cannot fully distinguish between the 3 variants tested as expected.

However the question of the false positive which is essential in a hyper-sensitive technique has claimed by the authors, remains an issue and the increase in the control sample numbers is just minimal while 2 unexplained some false positive persist

Finally the clinical relevance of this highly sensitive assay is not demonstrated in the vaccinated KT cohort as this assay is not a game changer in this application since there is no correlates of protection. To this reviewer's opinion the only clinical benefit of this assay would be in the very early antibody detection during the first days after infection

Reviewer #2 (Remarks to the Author):

na

Reviewer #3 (Remarks to the Author):

The authors have fully addressed by comments and suggestions

Reviewer #4 (Remarks to the Author):

The authors have adequately addressed most questions. But I have a few more comments:

- Abstract: should stress that further clinical evaluation is required to ascertain the performance of this novel assay
- What is the manufacturer of the CLIA used in this study? What is the cutoff value?
- Line 223-225 and Figure 3g: I do not understand how the results from IgG and IgM assay can be combined. If the authors want to compare the total antibody level between the CLIA test and their UCAD test, they should use a secondary antibody that can detect all antibody subtypes in their UCAD test. Please note that total antibody would also include IgA. Furthermore, if the IgM is positive and IgG is negative, would that be considered as positive or negative?

Reviewer #5 (Remarks to the Author):

I thank the authors for the impressive amount of additional work performed during the revision of their MS and for having adequately addressed my comments.

I only have two minor suggestions :

- 1) figure 5 and the effort to develop a point of care method by using lateral flow assay shouldn't be first cited in the discussion section but should instead be an integral part of the result section. please consider adding a paragraph presenting these data in the main body of the article.
- 2) Again, I don't really see what the cytometry data (presented in "S3. UCAD and cellular analysis of KTRs" in the Supplementary Information) bring to the present story. Since they are neither really new nor important for the message of this (already massive) article, I would consider removing them from the final version.

Response Letter

We thank all reviewers for their insightful comments and suggestions. We have revised manuscript accordingly. A point-by-point response letter is provided as follows.

Reviewer #1

Authors have answered adequately the major concerns raised by the first version and this revised manuscript has been strongly improved. The addition of more specificity investigations such as against the SARS-COV2 variants is promising although the UCAD cannot fully distinguish between the 3 variants tested as expected.

Comment-1: However, the question of the false positive which is essential in a hyper-sensitive technique has claimed by the authors, remains an issue and the increase in the control sample numbers is just minimal while 2 unexplained some false positive persist.

Response-1: We thank the reviewer for raising this concern. We have revised the manuscript by correcting description of the two “false positives” into two instances where UCAD and CLIA were not in agreement. As both samples were in the cohort of 55 negative sera collected during pandemic, it is not possible to know whether they were “false positives” or weak positive samples that were not detectable by CLIA.

Comment-2: Finally, the clinical relevance of this highly sensitive assay is not demonstrated in the vaccinated KT cohort as this assay is not a game changer in this application since there is no correlates of protection. To this reviewer's opinion the only clinical benefit of this assay would be in the very early antibody detection during the first days after infection.

Response-2: We thank the reviewer for raising the concern of clinical benefit of the assay. We have revised the abstract of the manuscript by toning down the claims related to clinical utility.

Reviewer #4

The authors have adequately addressed most questions. But I have a few more comments:

Comment-1: Abstract: should stress that further clinical evaluation is required to ascertain the performance of this novel assay.

Response-1: Agree. We have thoroughly revised the abstract to tone down the claims related to the clinical utility and stress the need for further clinical evaluation.

Comment-2: What is the manufacturer of the CLIA used in this study? What is the

cutoff value?

Response-2: The manufacturer of CLIA was Xiamen InnoDx Biotechnology CO. LTD and the cutoff value is 1.0. We added this information to the experimental section of the revised manuscript.

Comment-3: Line 223-225 and Figure 3g: I do not understand how the results from IgG and IgM assay can be combined. If the authors want to compare the total antibody level between the CLIA test and their UCAD test, they should use a secondary antibody that can detect all antibody subtypes in their UCAD test. Please note that total antibody would also include IgA. Furthermore, if the IgM is positive and IgG is negative, would that be considered as positive or negative?

Response-3: We apologize for the confusion. Here, we classify either IgM or IgG positive as a positive instance determined by UCAD and compared the results to the total antibody test by the standard CLIA. We revised the Figure 3g and the associate content in the manuscript to clarify how the positive was defined.

Reviewer #5

I thank the authors for the impressive amount of additional work perform during the revision of their MS and for having adequately addressed my comments. I only have two minor suggestions:

Comment-1: figure 5 and the effort to develop a point of care method by using lateral flow assay shouldn't be first cited in the discussion section but should instead be an integral part of the result section. Please consider adding a paragraph presenting these data in the main body of the article.

Response-1: We thank the reviewer for this suggestion, and we have revised manuscript by adding a new section on "Integrating UCAD with lateral flow readout" and moving Figure 5 to this new section.

Comment-2: Again, I don't really see what the cytometry data (presented in "S3. UCAD and cellular analysis of KTRs" in the Supplementary Information) bring to the present story. Since they are neither really new nor important for the message of this (already massive) article, I would consider removing them from the final version.

Response-2: The flow cytometry data shows significant differences in the levels of plasmablast, Th2, and Th17 between UCAD determined positive and negative groups in KTRs, so we believe that it is still meaningful to keep these data for readers who are interested in correlating humoral and cellular immunity upon COVID-19 infection or vaccination.